**Observation and analysis of spatio-temporal characteristics of surface ozone and carbon monoxide at multiple sites in the Kathmandu Valley, Nepal**

Khadak Singh Mahata[1,2], Maheswar Rupakheti[1,3], Arnico Kumar Panday [4,5], Piyush Bhardwaj[6], Manish Naja[6], Ashish Singh[1], Andrea Mues[1], Paolo Cristofanelli[7], Deepak Pudasainee[8], Paolo Bonasoni[7], Mark G. Lawrence[1,2]

[1]Institute for Advanced Sustainability Studies (IASS), Potsdam, Germany

[2]University of Potsdam, Potsdam, Germany

[3]Himalayan Sustainability Institute (HIMSI), Kathmandu, Nepal

[4]International Centre for Integrated Mountain Development (ICIMOD), Lalitpur, Nepal

[5]University of Virginia, Charlottesville, USA

[6]Aryabhatta Research Institute of Observational Sciences (ARIES), Nainital, India

[7]CNR-ISAC, National Research Council of Italy – Institute of Atmospheric Sciences and Climate, Bologna, Italy

[8]Department of Chemical and Materials Engineering, University of Alberta, Edmonton, Canada

Correspondence to: Maheswar Rupakheti (maheswar.rupakheti@iass-potsdam.de) and Khadak Singh Mahata (khadak.mahata@iass-potsdam.de)

## Abstract

Residents of the Kathmandu Valley experience severe particulate and gaseous air pollution throughout most of the year, even during much of the rainy season. The knowledge base for understanding the air pollution in the Kathmandu Valley was previously very limited, but is improving rapidly due to several field measurement studies conducted in the last few years. Thus far, most analyses of observations in the Kathmandu Valley have been limited to short periods of time at single locations. This study extends the past studies by examining the spatial and temporal characteristics of two important gaseous air pollutants (CO and $O_3$) based on simultaneous observations over a longer period at five locations within the valley and on its rim, including a supersite (at Bode in the valley center, 1345 m above sea level) and four satellite sites (at Paknajol, 1380 m asl in the Kathmandu city center, at Bhimdhunga (1522 m asl), a mountain pass on the valley's western rim, at Nagarkot (1901 m asl), another mountain pass on the eastern rim, and Naikhandi (1233 m asl), near the valley's only river outlet). CO and $O_3$ mixing ratios were monitored from January to July 2013, along with other gases and aerosol particles by instruments deployed at the Bode supersite during the international air pollution measurement campaign SusKat-ABC (Sustainable Atmosphere for the Kathmandu Valley – endorsed by the Atmospheric Brown Clouds program of UNEP). The monitoring of $O_3$ at Bode, Paknajol and Nagarkot as well as the CO monitoring at Bode were extended until March 2014 to investigate their variability over a complete annual cycle. Higher CO mixing ratios were found at Bode than at the outskirt sites (Bhimdhunga, Naikhandi and Nagarkot), and all sites except Nagarkot showed distinct diurnal cycles of CO mixing ratio with morning peaks and daytime lows. Seasonally, CO was higher during pre-monsoon (March-May) season and winter (December-February) season than during monsoon season (June-September) and post-monsoon (October-November) season. This is primarily due to the emissions from brick industries, which are only operational during this period (January-April), as well as increased domestic heating during winter, and regional forest fires and agro-residue burning during the pre-monsoon season. It was lower during the monsoon due to rainfall, which reduces open burning activities within the valley and in the surrounding regions, and thus reduces sources of CO. The meteorology of the valley also played a key role in determining the CO mixing ratios. The wind is calm and easterly in the shallow mixing layer, with a mixing layer height (MLH) of about 250 m, during the night

and early morning. The MLH slowly increases after the sunrises and decreases in the afternoon.
As a result, the westerly wind becomes active and reduces the mixing ratio during the day time.
Furthermore, there was evidence of an increase in the $O_3$ mixing ratios in the Kathmandu Valley
as a result of emissions in the Indo-Gangetic Plains (IGP) region, particularly from biomass
burning including agro-residue burning. A top-down estimate of the CO emission flux was made
by using the CO mixing ratio and mixing layer height measured at Bode. The estimated annual
CO flux at Bode was 4.9 µg m$^{-2}$ s$^{-1}$, which is 2-14 times higher than that in widely used emission
inventory databases (EDGAR HTAP, REAS and INTEX-B). This difference in CO flux between
Bode and other emission databases likely arises from large uncertainties in both the top-down
and bottom-up approaches to estimating the emission flux. The $O_3$ mixing ratio was found to be
highest during the pre-monsoon season at all sites, while the timing of the seasonal minimum
varied across the sites. The daily maximum 8 hour average $O_3$ exceeded the WHO recommended
guideline of 50 ppb on more days at the hilltop station of Nagarkot (159/357 days) than at the
urban valley bottom sites of Paknajol (132/354 days) and Bode (102/353 days), presumably due
to the influence of free-tropospheric air at the high-altitude site, as also indicated by Putero et al.,
(2015) for the Paknajol site in the Kathmandu Valley as well as to titration of $O_3$ by fresh NOx
emissions near the urban sites. More than 78% of the exceedance days were during the pre-
monsoon period at all sites. The high $O_3$ mixing ratio observed during the pre-monsoon period is
of a concern for human health and ecosystems, including agroecosystems in the Kathmandu
Valley and surrounding regions.

## 1. Introduction

Air pollution is one of the major health risks globally. It was responsible for premature loss of
about 7 million lives worldwide in 2012 (WHO, 2014), with about 1.7 million of these being in
South Asian countries (India, Pakistan, Nepal and Bangladesh) in 2013 (Forouzanfar, 2015). The
latest report shows that the indoor and outdoor air pollution are each responsible for 4 million
premature deaths every year (http://www.who.int/airpollution/en/). South Asia is considered to
be a major air pollution hotspot (Monks et al., 2009) and it is expected to be one of the most
polluted regions in the world for surface ozone ($O_3$) and other pollutants by 2030 (Dentener et
al., 2006; IEA 2016; OECD 2016). Past studies have shown that the air pollution from this
region affects not only the region itself, but is also transported to other parts of the world,
including comparatively pristine regions such as the Himalayas and the Tibetan plateau
(Bonasoni et al., 2010; Ming, et al., 2010; Lüthi et al., 2015), as well as to other distant locations
such as northern Africa and the Mediterranean (Lawrence and Lelieveld, 2010).The pollutants
are also uplifted to the tropopause by convective air masses and transported to the extratropical
stratosphere during the monsoon season (Tissier and Legras., 2016; Lawrence and Lelieveld,
2010; Fueglistaler et al., 2009; Highwood and Hoskins, 1998). Air pollution is particularly
alarming in many urban areas of South Asia, including in the city of Kathmandu and the broader
Kathmandu Valley, Nepal (Chen et al., 2015; Putero et al., 2015; Kim et al., 2015; Sarkar et al.,
2016; Shakya et al., 2017). This is due to their rapid urbanization, economic growth and the use
of poor technologies in the transportation, energy and industrial sectors. In Kathmandu
topography also plays a major role: the bowl-shaped Kathmandu Valley is surrounded by tall
mountains and only a handful of passes. Topography is a key factor in governing local
circulations, where low MLH  (typically in the range 250 m to 1,500 m)  and calm winds, have
been observed particularly during nights and mornings. This in turn results in poor ventilation
(Mues et al., 2017). Overall, this is conducive to trapping air pollutants and the deterioration of
air quality in the valley. Effectively mitigating air pollutants in the regions like the Kathmandu
Valley requires scientific knowledge about characteristics and sources of the pollutants. To
contribute to this urgently-needed scientific knowledge base, in this study we focus on the
analysis of measurements of two important gaseous species, carbon monoxide (CO and $O_3$, at
multiple sites in and around the Kathmandu Valley. This study analyzes data from January 2013
to March 2014, which includes the intensive phase of an international air pollution measurement
campaign – SusKat-ABC (Sustainable Atmosphere for the Kathmandu Valley – Atmospheric
Brown Clouds) – conducted during December 2012 - June 2013 (Rupakheti et al., 2018,
manuscript in preparation, submission anticipated in 1-2 months), with measurements of $O_3$ and
CO at some sites continuing beyond the intensive campaign period (Bhardwaj et al., 2017;
Mahata et al., 2017).
CO is a useful tracer of urban air pollution as it is primarily released during incomplete
combustion processes that are common in urban areas. Forest fires and agro-residue burning in
the IGP and foothills of the Himalaya are other important contributors of CO in the region
(Mahata et al., 2017; Bhardwaj et al., 2017). CO is toxic at high concentrations indoors and
outdoors, but our focus here is on ambient levels. The main anthropogenic sources of CO in the
Kathmandu Valley are vehicles, cooking activities (using liquefied petroleum gas, kerosene, and
firewood), and industries, including brick kilns, especially biomass co-fired kilns with older
technologies, and until recently diesel power generator sets (Panday and Prinn, 2009; Kim et al,
2015; Sarkar et al., 2016; Mahata et al., 2017; Sarkar et al., 2017). Tropospheric $O_3$, which is
formed by photochemical reactions involving oxides of nitrogen ($NO_x$) and volatile organic
compounds (VOCs), is a strong oxidizing agent in the troposphere. Because of its oxidizing
nature, it is also deleterious to human health and plants already at typically polluted ambient
levels (Lim et al., 2012; Burney and Ramanathan, 2014; Feng, 2015; Monks et al., 2015).
Tropospheric $O_3$ is estimated to be responsible for about 5-20 % of premature deaths caused by
air pollution globally (Brauer et al., 2012; Lim et al., 2012; Silva et al., 2013). It has also been
estimated that high concentrations of $O_3$ are responsible for a global loss of crops equivalent to $
11-18 billion annually (Avnery et al., 2011; UNEP and WMO, 2011), a substantial fraction of
which is associated with the loss in wheat in India alone (equivalent to $ 5 billion in 2010)
(Burney and Ramanathan, 2014). $O_3$ can also serve as a good indicator of the timing of the
breakup of the nighttime stable boundary layer (when the ozone levels increase rapidly in the
morning due to downward transport from the free troposphere (Panday and Prinn, 2009; Geiß et
al., 2017).
Only a few past studies have reported measurements of ambient CO mixing ratios in the
Kathmandu Valley. Davidson et al. (1986) measured CO in the city center and found mixing
ratios between 1 and 2.5 ppm in the winter (December – February) of 1982-1983. Panday and
Prinn (2009) measured similar levels of CO mixing ratios during September 2004 – June 2005,
although the main sources of CO shifted from biofuel-dominated air pollutants from cooking
activities in the 1980s to vehicle-dominated pollutants in the 2000s. The growth rate in the
vehicle fleet has had a substantial influence on air pollution in the valley, including CO and $O_3$.
Out of 2.33 million vehicles in Nepal, close to half of them are in the Kathmandu Valley (DoTM,
2015). Shrestha et al. (2013) estimated  annual emission of CO of 31 kt in 2010 from a fraction
of today's vehicle fleet in the Kathmandu Valley by using data from a field survey as input to the
International Vehicle Emission (IVE) model. The model simulation considered motorcycles,
buses, taxis, vans and three wheelers, but did not include personal cars, trucks and non-road
vehicles. The studied fleets covered ~73% of the total fleet (570,145) registered in the valley in
2010, with motorcycles being the most common vehicle (69% of the total fleet).
Past studies have investigated the diurnal and seasonal variations of CO and $O_3$ mixing ratios in
the Kathmandu Valley. Panday and Prinn (2009) observed distinct diurnal variations of CO
mixing ratios and particulate matter concentrations observed during September 2004 – June 2005
at Bouddha (about 4 km northwest of the SusKat-ABC supersite at Bode), with morning and
evening peaks. They found for the Kathmandu Valley that such peaks were created by the
interplay between the ventilation, as determined by the local meteorology, and the timing of
emissions, especially traffic and cooking emissions. The morning CO peak was also associated
with the recirculation of the pollutants transported down from an elevated residual pollution
layer (Panday and Prinn, 2009).
$O_3$ was observed to have lower nighttime levels in the city center than at the nearby hilltop site of
Nagarkot (Panday and Prinn, 2009). Pudasainee et al. (2006) studied the seasonal variations of
$O_3$ mixing ratios based on the observation for a whole year (2003-2004) in Pulchowk in the
Lalitpur district, just south of central Kathmandu Metropolitan City (KMC) in the Kathmandu
Valley. They reported seasonal $O_3$ mixing ratios to be highest during the pre-monsoon (March –
May) and lowest in the winter (December – February). As a part of the SusKat-ABC Campaign,
Putero et al. (2015) monitored $O_3$ mixing ratios at Paknajol, an urban site in the center of the
KMC over a full-year period (February 2013-January 2014). They also observed similar seasonal
variations in $O_3$ mixing ratios in the valley to those observed by Pudasainee et al. (2006), with
highest $O_3$ during the pre-monsoon (1 February – 12 May) season, followed by the monsoon (13
May – 6 October), post-monsoon (7 October – 26 October) and winter (27 October – 31 January)
seasons. They found that during the pre-monsoon season, westerly winds and regional synoptic
circulation transport $O_3$ and its precursors from regional forest fires located outside the
Kathmandu Valley. In another study conducted as part of the SusKat-ABC Campaign, 37 non-
methane volatile organic compounds (NMVOCs) were measured at Bode, with data recording
every second, during winter of 2012-2013; the measurements included isoprene, an important
biogenic precursor of $O_3$ (Sarkar et al., 2016). They found concentrations to vary in two distinct
periods. The first period was marked by no brick kiln operations and was associated with high
biogenic emissions of isoprene. During the second period nearby brick kilns, which use coal
mixed with biomass, were in; they contributed to elevated concentrations of ambient acetonitrile,
benzene and isocyanic acid. Furthermore, the authors found that oxygenated NMVOCs and
isoprene combined accounted for 72% and 68% of the total $O_3$ production potential in the first
period and second period, respectively.
Prior to the SusKat-ABC campaign there were no studies that simultaneously measured ambient
CO and $O_3$ mixing ratios at multiple sites in the Kathmandu Valley over extended periods of
time. Past studies either focused on one long-term site, or on short-term observation records at
various sites (Panday and Prinn, 2009), or they investigated the seasonal characteristics of single
pollutants such as $O_3$ at a single site in the valley (Pudasainee et al., 2006). The most comparable
past study is by Putero et al. (2015), who described $O_3$ mixing ratios at one SusKat-ABC site
(Paknajol) in the Kathmandu city center observed during the SusKat-ABC campaign, and
discussed $O_3$ seasonal variations. There is also a companion study on regional CO and $O_3$
pollution by Bhardwaj et al. (2017) which is based on $O_3$ and CO mixing ratios monitored at the
SusKat-ABC supersite at Bode in the Kathmandu Valley for a limited period (January-June
2013) and at two sites in India (Pantnagar in Indo-Gangetic Plain and Nainital in Himalayan
foothill). They reported simultaneous enhancement in $O_3$ and CO levels at these three sites in
spring, highlighting contribution of regional emissions, such as biomass burning in northwest
Indo-Gangetic Plain (IGP), and regional transport to broader regional scale pollution, including
in the Kathmandu Valley. In this study, we document the diurnal and seasonal (where applicable)
characteristics and spatial distributions of CO and $O_3$ mixing ratios based on simultaneous
observations at several locations within the valley and on the valley rim mountains over a full
year, helping to characterize the pollution within the valley and the pollution plume entering and
exiting the valley. We also compute the first top-down estimates of CO emission fluxes for the
Kathmandu Valley and compare these to CO emissions fluxes in widely-used emission datasets
such as EDGAR HTAP (Janssens-Maenhout et al., 2000), REAS (Kurokawa et al., 2013) and
INTEX-B (Zhang et al., 2009).

**2. Study sites and methods**
The Kathmandu Valley, situated in the foothills of the Central Himalaya, is home to more than 3
million people. The valley floor has an area of about 340 km$^2$, with an average altitude of about
1300 m above sea level (m asl). It is surrounded by peaks of about 1900-2800 m asl. The valley
has five major mountain passes on its rim: the Nagdhunga, Bhimdhunga and Mudku Bhanjhyang
passes in the west, and the Nala and Nagarkot passes in the east, as shown in Figure 1. The
passes are situated at altitudes of 1480-1530 m asl. There is also one river outlet (the Bagmati
River) towards the south, which constitutes a sixth pass for air circulation in and out of the valley
(Regmi et al., 2003; Panday and Prinn, 2009). We selected five measurement sites, including two
on the valley floor (Bode and Paknajol), two on mountain ridges (Bhimdhunga and Nagarkot)
and one near the Bagmati River outlet (Naikhandi) to characterize the spatial and temporal
variabilities of CO and $O_3$ mixing ratios in the Kathmandu Valley. A short description of the
measurement sites is presented here and in Table 1, while details on instruments deployed at
those sites for this study are presented in Table 2. Further details of the measurement sites are
described in the SusKat-ABC campaign overview paper (Rupakheti et al., 2017, manuscript in
preparation).

Bode (27.69°N and 85.40°E, 1344 m asl): This was the supersite of the SusKat-ABC Campaign.
Bode is located in the Madhyapur Thimi municipality in the just east of the geographic center of
the valley. It is a semi-urban site surrounded by urban buildings and residential houses scattered
across agricultural lands. Within 4 km there are 10 brick kilns and the Bhaktapur Industrial
Estate towards the southeast (refer to Sarkar et al., 2016; Mahata et al., 2017 for details). The $O_3$
and CO instruments at Bode site were placed on the fifth floor of a 6-story building, the tallest in
the area. The inlets of the CO and $O_3$ analyzers were mounted on the roof top of the temporary
lab, 20 m above the ground level.

Bhimdhunga: This site (27.73°N, 85.23°E, 1522 m asl) is located on the Bhimdhunga pass on the
western rim of the valley. It is one of the lowest points on the north-south running mountain
ridge between the Kathmandu Valley to the east and a valley of a tributary of the Trishuli River
to the west. It is situated about 5.5 km from the western edge of the KMC (Kathmandu
Metropolitan City), in a rural setting with very few scattered rural houses nearby. The CO
instrument was placed on the ground floor of a small one-story building and its inlet was 2 m
above ground. An automatic weather station (AWS) (Hobo Onset, USA) was set up on the roof
of another one-story building at a distance of ca. 15 m from the first building.

Paknajol: This site (27.72°N, 85.30°E, 1380 m asl) is located at the city center in the KMC, near
the popular touristic area of Thamel. It is in the western part of the valley and about 10 km
distance from the Bode supersite. The $O_3$ and meteorological instruments relevant to this study
were placed on the top floor and rooftop of a 6-story building, the tallest in the area (details in
Putero et al., 2015; note that CO was not measured here). The inlet of the $O_3$ analyzer was placed
25 m above the ground.

Naikhandi: This site (27.60°N, 85.29°E, 1233 m asl) is located within the premises of a school
(Kamdhenu Madhyamik Vidhyalaya) located at the southwestern part of the valley (~7 km south
from the nearest point of the Ring Road). The school premise is open, surrounded by sparsely
scattered rural houses in agricultural lands. The nearest village (~75 houses) is about 500 m away
in the southwest direction. There are 5 brick kilns within 2 km distance (2 to the north and 3 to
the northeast) from the site. The instruments were kept in a one-story building of the school and
its inlet was 5 m above the ground. The AWS (Hobo Onset, USA) was installed on the ground
near the Bagmati River, ~100 m away from the main measurement site.

Nagarkot: This site is located on a mountain ridge (27.72°N, 85.52°E, 1901 m asl), ca. 13 km
east of Bode, in the eastern part of the valley. The site faces the Kathmandu Valley to the west
and small rural town, Nagarkot, to the east. The instruments were set up in a 2-story building of
the Nagarkot Health Post and their inlets were 5 m above the ground. The AWS (Vaisala
WXT520, Finland) was set up on the roof of the building.

Bhimdhunga Pass in the west and Naikhandi near the Bagmati River outlet in the southwest are
the important places for interchange of valley air with outside air. The Bhimdhunga and
Naikhandi sites are approximately 5.5 and 7 km away from the nearest edge of the city,
respectively. Similarly, Bode is located downwind of the city centers and thus receives pollution
outflow from nearby city centers of Kathmandu/Lalitpur due to strong westerly and
southwesterly winds (4-6 m s$^{-1}$) during the day time, and emissions from the Bhaktapur area to
the east and southeast direction by calm easterly winds (< 1 m s$^{-1}$) during the night (Sarkar et al.,
2016; Mahata et al., 2017).

A freshly calibrated new CO analyzer (Horiba APMA-370, Japan) was deployed for the first
time at Bode. This instrument is based on the IR absorption method at 4.6 µm by CO molecules.
Before field deployment at Bode, it was compared with the bench model of the Horiba (APMA-
370), and the correlation (r) between them was 0.9 and slope was 1.09. The instrument was
regularly maintained by running auto-zero checks (Bhardwaj et al., 2017). Similarly, another CO
analyzer (Picarro G2401, USA) which is based on cavity ring-down spectroscopy technique
(CRDS) was also a new factory calibrated unit, and was deployed in Bode along with the Horiba
APMA-370. An IR-based Thermo CO monitor (model 48i-TLE) was run simultaneously with a
co-located cavity ring down spectrometry based Picarro CO analyzer for nearly 3 months. The
correlation coefficient and slope between the two measurements were found to be 0.99 and 0.96,
respectively (Mahata et al., 2017). This indicates that there was very little drift in the IR-based
CO values due to room temperature change, within acceptable range (i.e., within the
measurement uncertainties of the instruments). Therefore, we did not any apply correction in the
IR-based CO data. All other CO analyzers (Thermo Scientific, 48i-TLE, USA), which are also
based on IR absorption by CO molecules, deployed at Bhimdhunga, Naikhandi and Nagarkot,
were set up for regular automatic zero checks on a daily basis. In addition, a span check was also
performed during the observations by using span gas of 1.99 ppm (Gts-Welco, PA, USA) on
March 8, 2013 at Naikhandi and Nagarkot, and on March 9 at Bhimdhunga. The IR-based CO
instruments' span drifts were within a 5 % range.

For the $O_3$ monitor (Teledyne 400E, USA) at Bode, regular zero and span checks were carried
out using the built-in $O_3$ generator and scrubber (Bhardwaj et al., 2017). This unit was used in
Bode from 01 January 2013 to 09 June 2013. New factory-calibrated $O_3$ monitors (Thermo
Scientific, 49i, USA) were used for the rest of the measurement period (18 June 2013 to 31
December 2013) at Bode, and for the full year of measurements at Nagarkot. A Thermo
Environmental $O_3$ analyzer (Model 49i, USA) was used at the Paknajol site (Putero et al., 2015)
with the same experimental set up as described in Cristofanelli et al. (2010). The working
principle of all of the $O_3$ instruments is based on the attenuation of UV radiation by $O_3$ molecules
at ~254 nm.
In order to characterize observations across the seasons, we considered the following seasons as
defined in Shrestha et al. (1999) and used in other previous studies in the Kathmandu Valley
(Sharma et al., 2012; Chen et al., 2016; Mahata et al, 2017): Pre-monsoon (March, April, May);
Monsoon (June, July, August September); Post-monsoon (October, November); and Winter
(December, January, February).

**3. Results and discussion**
**3.1 CO mixing ratio at multiple sites**
Figure 2 shows the time series of the hourly average CO mixing ratios at the four sites (Bode,
Bhimdhunga, Naikhandi and Nagarkot). Fluctuations in CO mixing ratios were higher during the
winter and pre-monsoon than during the monsoon season at all sites. The monsoon rain generally
starts in Nepal around mid-June. In 2013, however, there were more frequent rain events in the
month of May than in previous years. The CO mixing ratios (measured in parts per billion by
volume, hereafter the unit is denoted as ppb) of hourly averaged data over the total observation
periods at four sites and their standard deviation were: Bode (569.9 ± 383.5) ppb during 1
January - 15 July, Bhimdhunga (321.5 ± 166.2) ppb during 14 Jan - 15 July, Naikhandi (345.4 ±
147.9) ppb during 3 January - 6 June and Nagarkot (235.5 ± 106.2) ppb during 13 February - 15
July (except 4 April to 7 June). Nagarkot had only about 3 months of CO data (due to a problem
in zeroing of the instrument) during the observation period. For the measurement period, the CO
mixing ratio at Nagarkot (~13 km far from Bode) showed small fluctuations compared with the
other sites. High CO values in the Kathmandu Valley during the dry season (November-May)
were also reported by Panday and Prinn (2009) based on their measurements during September
2004-May 2005 at Bouddha (~ 4 km in northwest from Bode). The simultaneous episodes of
high CO observed from April 1to15 in Bhimdhunga, Bode and Naikhandi indicate the influence
of regional sources, in addition to local sources. This is discussed further in section 3.2.3.

**3.2  Diurnal and seasonal variations of CO**
**3.2.1   Diurnal pattern of CO at multiple sites**
Figure 3 shows the diurnal cycles of CO mixing ratios at four sites (plotted for the period of 13
February to 3 April 2013, when the data were available from all four sites). The variation in the
mixing ratios during the day was characterized by a pronounced morning peak, a weaker evening
peak, and a daytime low; except at Nagarkot where peaks are less visible. Multiple sources
contribute to the morning and evening peaks, especially emission from vehicles, residential
burning (fossil fuel and biomass), brick kilns and trash burning (Kim et al., 2015; Sarkar et al.,
2016; Mahata et al., 2017). Other studies conducted during the SusKat-ABC campaign have
identified garbage (household waste and yard waste) burning as a key source of various air
pollutants, such as OC and EC (Kim et al., 2015), PAHs (Chen et al., 2015), and NMVOCs
(Sarkar et al., 2016; Sarkar et al., 2017). Garbage burning is often done in small fires and quite
sporadic, normally taking place in the evenings and mornings (partly chosen to avoid attention
from the responsible authorities). The rate of waste (and also biomass) burning in the morning is
higher in winter due to the use of the fires for providing warmth on colder days.
The observed diurnal cycle of CO is similar to that reported in a previous study (Panday and
Prinn, 2009), and is also similar to the diurnal pattern of black carbon (BC) in the Kathmandu
Valley (Sharma et al., 2012; Mues et al., 2017). The diurnal cycles of these primary pollutants
are closely coupled with the valley's boundary layer height, which is about 1200 m during
daytime, and falls to approximately 200 m at nighttime in Bode (Mues et al., 2017). Nagarkot
and Bhimdhunga, both on mountain ridges, are generally above the valley's boundary layer,
especially at night, and thus the diurnal profile especially at Nagarkot is distinct compared to
other three sites, being relatively flat with small dip during 12:00-18:00.

Distinct morning peaks were observed in Bode, Bhimdhunga and Naikhandi at 08:00, 09:00, and
10:00, respectively, i.e., the morning peak lags by 1-2 hours in Bhimdhunga and Naikhandi
compared to Bode. Bhimdhunga on the mountain ridge may receive the Kathmandu Valley's
pollution due to upslope winds (~2 m s$^{-1}$) from the east direction in the morning hours after the
dissolution of the valley's boundary layer due to radiative heating of the mountain slopes. On the
other hand, Naikhandi is in close proximity to brick kilns and could be impacted by their plumes
carried to the site by northerly winds in the early morning (ca. 07:00-10:00, not shown). The
evening peak values at Bode and Bhimdhunga were less pronounced compared to the morning
maxima. The morning peak at Bode was influenced by nighttime accumulation of CO along with
other pollutants from nearby brick kilns (Sarkar et al., 2016; Mahata et al., 2017; Mues et al.,
2017) and recirculation of air from above (Panday and Prinn, 2009). Similarly, the local
pollution from the nearby village and city area due to upslope winds from the valley floor is
expected to contribute to the morning peak at Bhimdhunga. The evening peak at Naikhandi was

at 21:00 and was closer to the morning values in comparison to the large difference between morning and evening peaks at Bode and Bhimdhunga. A nighttime build-up (linear increase) of various pollutants compared to the afternoon minimum was typically observed in Bode during the SusKat-ABC measurement period, including the main campaign period (Sarkar et al., 2016; Mahata et al., 2017; Mues et al., 2017). This is mainly associated with the persistent emissions such as those from brick kilns, which are in close proximity to the Bode measurement site under the stable boundary layer. The isolated peak during the morning transition phase at Bhimdhunga could be due to an elevated polluted layer because of the slope wind (Panday et al., 2009). There appears to be less influence of nighttime polluting sources at Naikhandi and Bhimdhunga than at Bode.

The MLH starts increasing after radiative heating of the surface by incoming solar radiation. The heating of the ground causes thermals to rise from the surface layer resulting in the entrainment of cleaner air from above the boundary layer leading to the dissolution of nocturnal stable boundary layer. Increasing wind speeds (4-6 m s$^{-1}$) during daytime also support turbulent vertical diffusion, as well as flushing of the pollution by less polluted air masses from outside the valley, with stronger horizontal winds allowing significant transport of air masses into the valley. In addition, reduced traffic and household cooking activities during daytime compared to morning and evening rush hours contribute to the reduced CO mixing ratios.

**3.2.2 CO diurnal variation across seasons**

Due to the lack of availability of simultaneous CO data at all sites covering the entire sampling period, a one-month period was selected for each season to examine the diurnal variation across the seasons, and to get more insights into the mixing ratios at different times of the day, as reported in Table 4. Figure 4 shows the diurnal variation of CO mixing ratios in Bode, Bhimdhunga, and Naikhandi during the selected periods for the three seasons.

The diurnal cycles during each season had different characteristics. The most prominent distinction was that the CO mixing ratio was low during the monsoon period over all sites (Figure 4, Table 4) as a result of summer monsoon rainfall in the valley, which is 60 - 90% of the 1400 mm rainfall for a typical year (Nayava, 1980; Giri et al., 2006). Because of the rainfall, the

brick production activities are stopped in the valley (usually they are operational from January-
April every year). Further, the rainfall also diminishes many burning activities (forest fires, agro-
residue and trash burning) within the valley and surrounding region, and thus reduces CO
emissions. Afternoon CO mixing ratios were higher in the pre-monsoon season than in the other
two seasons in Bode, Bhimdhunga and Naikhandi (also see Table 4), with the most likely
sources being emissions from forest fires and agro-residue burning arriving from outside the
valley during this season (this will be discussed further in section 3.2.3). Nighttime accumulation
was observed in Bode and Bhimdhunga, but not at Naikhandi, where the mixing ratio decreased
slightly from about 20:00 until about 04:00, after which the mixing ratios increased until the
morning peak. The nighttime accumulation of CO in Bode during pre-monsoon and winter is due
to the influence of nearby brick kilns (Mahata et al., 2017) because of the calm easterly wind
(refer supplementary Figure S2 in Mahata et al., 2017). Previous studies carried out at the Bode
site during the SusKat-ABC campaign have attributed over a dozen brick kilns located near Bode
as strong sources of BC and EC (Kim et al., 2015; Mues et al., 2017), NMVOCs (Sarkar et al,
2016; Sarkar et al., 2017), SO2 (Kiros et al., 2016) and CO (Mahata et al., 2017), and the
enhanced concentrations were observed during nighttime and mornings when winds blew from
east and southeast bringing emissions from the location of the brick kilns to the observation site.
Bhimdhunga is not near any major polluting sources such as brick kilns, and it is unclear whether
the nighttime CO accumulation in Bhimdhunga is primarily due to ongoing local residential
pollution emissions, and/or to pollution transported from remote sources. The transition of the
wind from westerlies during the day to easterlies during the night, with moderate wind speed
($\sim$2-4 m s$^{-1}$) at Bhimdhunga, may bring polluted air masses westwards which were initially
transported to the eastern part from the Kathmandu Valley during the daytime (Regmi et al.,
2003; Panday and Prinn, 2009; Panday et al., 2009).
The distinct shift in the morning peak was seen at all 3 sites by season. The one hour shift in the
morning peak from the pre-monsoon to winter is due to an earlier onset of the morning
transition. However, the one hour difference in the morning peak between Bode (pre-monsoon at
8:00; winter at 9:00) and Bhimdhunga/Naikhandi (pre-monsoon at 9:00; winter at 10:00) in the
pre-monsoon and winter is associated with commencement of early local emissions under the

shallow boundary layer at Bode. The one hour lag in the morning peak at Bhimdhunga and Naikhandi may be due to transport of city pollutants to the site, respectively.

Across the seasons, the afternoon (12:00-16:00) CO mixing ratio was higher during the pre-monsoon than in the winter at all three stations (p value for all sites < 0.5) (Table 4), although the mixing layer was higher in the pre–monsoon season than in the winter in Bode (and presumably at the other sites as well). This is not likely to be explained by local emissions in the valley, since these are similar in the winter and pre-monsoon periods. Putero et al. (2015) suggested instead that this reflects an influx of polluted air into the valley due to large synoptic circulation patterns during the pre-monsoon season. Such regional influences are explored further in the next section.

### 3.2.3 Regional influence on CO in the valley

Recent studies have indicated the likelihood of regional long-range transport contributing to air pollution in different parts of Nepal (Marinoni et al., 2013; Tripathee et al., 2014; Dhungel et al., 2016; Rupakheti et al., 2016; Lüthi et al., 2016; Wan et al., 2017), including the Kathmandu Valley, especially during the pre-monsoon period (Panday and Prinn, 2009; Putero et al., 2015; Bhardwaj et al., 2017). During the pre-monsoon season, frequent agro-residue burning and forest fires are reported in the IGP region including southern Nepal and the Himalayan foothills in India and Nepal (Ram and Sarin, 2010; Vadrevu et al., 2012), and in the Kathmandu Valley. This season is also characterized by the strongest daytime local wind speeds (averaging 4-6 m s$^{-1}$) in the Kathmandu Valley (Mahata et al., 2017). Our study also observed several episodes of days with both elevated CO mixing ratios (Figure 2) and $O_3$ mixing ratios (also measured in parts per billion by volume, hereafter the unit is denoted as ppb) (Figure 5) during April and May, especially during the late afternoon period. The influence of regional pollutants was investigated by comparing a 2-week period with normal CO levels (16–30 March (hereafter "period I") with an adjacent two week period (1-15 April) with episodically high CO mixing ratios (hereafter "period II"), which nicely fit with the "burst" in regional fire activity presented by Putero et al. (2015) in their Figure 9. The t-test of the two hourly data means of CO in period I and period II at Bode, Bhimdhunga and Naikhandi (as in Figure 5) were performed at 95% confidence level and the differences were found to be statistically significant (p < 0.5).

Figure 5a shows the diurnal cycle of CO mixing ratios during period I (faint color) and period II
(dark color) at Bode, Bhimdhunga and Naikhandi. The difference between two periods was
calculated by subtracting the average of period I from average of period II. The average CO
mixing ratios during period II were elevated with respect to period I by 157 ppb at Bode, 175
ppb at Bhimdhunga, and 176 ppb at Naikhandi. The enhancements in mixing ratios at the three
sites were fairly similar from hour to hour throughout the day, with the exception of the late
afternoon when the enhancement was generally greatest. This consistency across the sites
suggests that the episode was caused by a large-scale enhancement (regional contribution) being
added onto the prevailing local pollution levels at all the sites. A large-scale source would also
be consistent with the greater enhancements of CO at the outskirt sites, which would be most
directly affected by regional pollution, compared to the central valley site of Bode, with strong
local sources. The enhancement during the period II is substantial (statistically significant as
mentioned above), representing an increase of approximately 45% at the outskirt sites of
Bhimdhunga and Naikhandi (which start with lower CO levels), and 23% at Bode. During both
periods I and II, local winds from west (the opposite direction from the brick kilns, which were
mostly located to the southeast of Bode) were dominant during daytime at Bode (Figure 5b, c).
This suggests that the elevation in CO levels was caused by additional emissions in period II in
the regions to the west and southwest of the Kathmandu Valley, e.g., large scale agricultural
burning and forest fires during this period, as also noted by Putero et al. (2015) (see their Figure
9). Far away, in Lumbini in the southern part of Nepal (Rupakheti et al., 2016), and Pantnagar in
northern IGP in India (Bharwdwaj et al., 2017), about 220 km (aerial distance) to the southwest
and 585 km to the west, respectively, of the Kathmandu Valley, CO episodes were also observed
during the spring season of 2013, providing a strong indication that the episode in period II was
indeed regional in nature.

**3.3 $O_3$ in the Kathmandu Valley and surrounding areas**
Figure 6 shows the hourly average and daily maximum 8-hour average of $O_3$ mixing ratios at
Bode, Paknajol, and Nagarkot from measurements during the SusKat campaign and afterwards,
along with $O_3$ mixing ratios from a previous study (November 2003 - October 2004; Pudasainee
et al., 2006) at the Pulchowk site (4 km away from Paknajol) in the Latitpur district. The daily
maximum 8-hour average $O_3$ was calculated by selecting the maximum $O_3$ mixing ratio from 8
hour running averages during each day. The nighttime mixing ratio of hourly $O_3$ drops close to
zero in Bode, Paknajol and Pulchowk in the winter season. This is a typical characteristic of
many urban areas where reaction with NO at night depletes $O_3$ from the boundary layer (e.g.,
Talbot et al., 2005). In the pre-monsoon and monsoon months, the titration is not as strong and
the hourly $O_3$ falls, but generally remains above 10 ppb. Similar patterns of ozone mixing ratios
were observed at other sites in northern South Asia. For example, higher $O_3$ mixing ratios were
observed in the afternoon (84 ppb) and lower during the night and early morning hours (10 ppb)
at Kullu Valley, a semi-urban site located at 1154 m asl, in the North-western Himalaya in India
(Sharma et al. 2012). A similar dip in $O_3$ value in the dark hours was observed at Ahmedabad,
India by Lal et al. (2000). Nagarkot, in contrast, is above the valley's boundary layer and has
lesser NO for titration at night at this hill station as has been observed in another hill station in
Himalayan foothills (Naja and Lal, 2002). Thus, the $O_3$ level remains above 25 ppb during the
entire year at Nagarkot. As also shown in Table 3, at all sites, the $O_3$ mixing ratios were highest
in the pre-monsoon, but the timing of the lowest seasonal values varied across the sites: post-
monsoon in Bode, winter in Paknajol and monsoon in Nagarkot. Such differences in minimum
$O_3$ across the sites can be anticipated due to the locations of the sites (e.g., urban, semi-urban,
rural and hilltop sites, with differing availabilities of $O_3$ precursors from different emission
sources). The seasonal variations of $O_3$ observed at Bode in this study are consistent with Putero
et al. (2015) and Pudasainee et al. (2006), who also observed $O_3$ maxima during the pre-
monsoon, but $O_3$ minima during the winter season.
The daily maximum 8-hour average $O_3$ mixing ratio (solid colored circles in Figure 6) exceeded
the WHO recommended guideline of 50 ppb (WHO, 2006, black dotted line in Figure 6) most
frequently during the pre-monsoon period and the winter. During the observation period, the
daily maximum 8-hour average $O_3$ exceeded the WHO guideline on 102 out of 353 days of
observation (29%) at Bode, 132/354 days (37%) at Paknajol and 159/357 days (45%) at
Nagarkot. The higher exceedance rate at Nagarkot is because it is at higher altitude, which
results in (i) greater exposure to large-scale regional pollution, especially from forest fire in the
Himalayan foothills and agro-residue burning in the IGP region, outside the Kathmandu Valley
(Sinha et al., 2014, Putero et al., 2015), (ii) less titration of $O_3$ by $NO_x$, being farther away from
the main pollution sources, and (iii) exposure to $O_3$ rich free tropospheric air, including
influences from stratospheric intrusions. The diurnal profiles of $O_3$ mixing ratios (Figure 7) at
three sites Bode and Pakanajol in the Valley and Nagarkot, a hilltop site normally above the
Kathmandu Valley's boundary layer shows, notably in the morning hours, that the residual layer
above the Kathmandu Valley's mixing layer contains a significant amount of ozone. Based on
the surface ozone data collected at Paknajol during 2013-14, Putero e al. (2015) concluded that
downward mixing of ozone from the residual layer contributes to surface ozone in the
Kathmandu Valley in the afternoon hours (11:00-17:00 local time). It is likely that the same
source has also contributed to higher ozone mixing ratios at Nagarkot. Such mixing has been
observed at other sites as well. Wang et al. (2012) reported the increase in downward mixing of
$O_3$ from the stratosphere to the middle troposphere (56%) and the lower troposphere (13%) in
spring and summer in Beijing. The downward flux was highest in the middle troposphere (75%)
in winter. Similarly, Kumar et al. (2010) reported that more than 10 ppb of stratospheric
contribution at a high altitude site (in Nainital) during January to April. However, there were no
significant stratospheric intrusions seen in spring and summer (seen only in winter) at Nepal
Climate Observatory-Pyramid (Putero et al., 2016).
During the SusKat-ABC campaign in 2013 and later in 2014, passive sampling of gaseous
pollutants ($SO_2$, $NO_x$, $NH_3$ and $O_3$) was carried out at fourteen sites including urban/semi-urban
sites (Bode, Indrachowk, Maharajganj, Mangal Bazar, Suryabinayak, Bhaisepati,
Budhanilkantha, Kirtipur, and Lubhu) and rural sites (Bhimdhunga, Naikhandi, Sankhu,
Tinpiple, and Nagarkot) in the Kathmandu Valley (Kiros et al., 2016). Similar to this study, they
also observed higher $O_3$ mixing ratios in rural areas than the urban/semi-urban sites in the
Kathmandu Valley. Exceedances of the WHO standard are most common during the pre-
monsoon season, occurring 78% (72/92 days), 88% (78/89 days) and 92% (85/92 days) of the
time at Bode, Paknajol and Nagarkot, respectively. A study by Putero et al., (2015), based on $O_3$
mixing ratio measurements at Paknajol in the Kathmandu Valley, as a part of the SusKat-ABC
campaign, has reported that the dynamics (both by horizontal and vertical winds) plays a key role
in increased $O_3$ mixing ratios in the afternoon in the Kathmandu Valley. They estimated that the
contribution of photochemistry varied as a function of the hour of the day, ranging from 6 to 34
%. Unfortunately, no viable NOx measurements were obtained at any site in the Kathmandu
Valley and surrounding mountain ridges during the SusKat-ABC campaign. Speciated VOCs
were measured at Bode only for about 2 months but NOx was not available for the same period.
Therefore we were not able to discern quantitatively proportional contributions of NOx, VOCs
and intrusion (chemistry vs. dynamics) from the free troposphere or lower stratosphere to
observed $O_3$ concentrations at Nagarkot, Bode and other sites in the Valley. In the context of
protecting public health, crops and regional vegetation, the $O_3$ mixing ratios in the Kathmandu
Valley and surrounding areas clearly indicate the urgent need for mitigation action aimed at
reducing emissions of its precursor gases NOx and VOCs. However, air quality management
plans need to consider carefully the reduction strategies of NMVOCs or NOx while aiming at
mitigating the $O_3$ pollution in the Kathmandu Valley. If the correct strategy (NMVOCs vs. NOx)
is not applied, then $O_3$ mixing ratios could increase, for example, as seen in Huszar et al. (2016)
where they reported that reducing NMVOCs in urban areas in central Europe leads to $O_3$
reduction whereas the focus on NOx reduction results in $O_3$ increase.

The SusKat-ABC $O_3$ data can be compared to observations made about a decade ago by
Pudasainee et al. (2006) at the urban site of Pulchowk, not far from Paknajol, as plotted in Figure
6d. The daily maximum 8-hour average $O_3$ had exceeded the WHO guideline at Pulchowk for
33% (95/292 days) of days during the observation from November 2003 to October 2004. The
exceedance was 38% (133/354 days) of days at Paknajol during Feb 2013 - March 2014. Due to
inter-annual variability and differences in the seasonal observation time periods at Pulchowk and
Pakanajol, we cannot draw any conclusions about trends over the decade between the
observations because of the difference in location and sampling height as well as a general
difference in instrument calibration. However, a clear similarity between the observations is that
most of the exceedance took place during pre-monsoon season, during which both studies have
observations throughout the season (~90 days). The percentage of exceedance at Pulchowk
during the pre-monsoon season in 2003-2004 was 70% (63/90 days) and at Pakanajol in 2013 it
was 88% (78/89 days). However, just like for the annual fraction of exceedances, due to inter-
annual variability we cannot say that the 18% (or ca. 15 days) difference in the exceedances is
significant. A longer term $O_3$ record would be needed to really establish if there is a trend in the
ozone concentrations.

**3.4 $O_3$ seasonal and diurnal variation**
The seasonal average $O_3$ mixing ratios at Bode, Nagarkot and Paknajol are shown in Table 3. For
comparison, the $O_3$ mixing ratios measured at two sites in India, (i) Manora Peak (1958 m asl),

ca. 9 km from Nainital city, a site in rural mountain setting and (ii) Delhi, a highly-polluted urban setting in northwest IGP are also listed in the Table, based on results from Kumar et al. (2010) and Ghude et al. (2008). There is a strong similarity between the urban and semi-urban sites in Nepal (i.e., Bode, Pakanajol) and India (i.e., Delhi), as well as between the rural and mountain sites in Nepal (i.e., Nagarkot) and India (i.e., Manora Peak), with small differences. The peak mixing ratios were in the pre-monsoon period: at the rural and mountain sites the peak ozone mixing ratio values were very similar (64 and 62 ppb for Nagarkot and Manora Peak, respectively) and are due to influences discussed earlier for Nagarkot; at the sub-urban and urban sites the pre-monsoon values are significantly lower (ca. 40, 42, 33 ppb for Bode, Paknajol, Delhi, respectively) due to fresh NOx emissions near the urban sites and the consequent titration of ozone with NO. The lowest $O_3$ seasonal values at rural and mountain sites typically occur in the monsoon months while for semi-urban and urban sites, the minimum was observed during post-monsoon (Bode) and winter (Paknajol).

Figure 7 shows the diurnal variation of $O_3$ mixing ratios at Bode, Paknajol and Nagarkot in the different seasons. The typical $O_3$ maximum mixing ratio in the early afternoon at the urban and semi-urban sites is mainly due to daytime photochemical production as well as entrainment of ozone due to dynamics (both intrusion of ozone rich free tropospheric air into the boundary layer, and regional scale horizontal transport of ozone), as explained in case of Paknajol by Putero et al. (2015).

The ozone mixing ratios are relatively constant throughout the day at Nagarkot (~1901 m asl), which, being a hilltop site, is largely representative of the lower free tropospheric regional pollution values, however, it is also affected by ozone production from precursors transported from the Kathmandu Valley due to westerly winds during the afternoon hours. The dip in $O_3$ at Nagarkot (Figure 7) in the morning transition hours indicates the upward mixing of air from polluted (and Ozone-depleted) nocturnal boundary layer as it is breaking up.

**3.5 CO emission flux estimate**

It is possible to determine a top-down estimate of the average CO emission flux for the morning hours for the region around the Bode site by applying an approach that was developed and used in Mues et al. (2017) to estimate the emission fluxes of BC at Bode. The analysis of Mues et al. (2017) found BC fluxes for the Kathmandu Valley that were considerably higher than the

widely-used EDGAR HTAP emission database (Version 2.2). Support for this top-down estimate
was found by considering the BC concentrations and fluxes for the Kathmandu Valley in
comparison to Delhi and Mumbai; although the observed BC concentrations were similar in all
three locations, the EDGAR HTAP V2.2 emissions of BC for the Kathmandu Valley are much
lower than those for Delhi and Mumbai, while the top-down emissions estimate for the
Kathmandu Valley were similar to the emissions from EDGAR HTAP V2.2 for Delhi and
Mumbai (Mues et al., 2017).

Here we apply the same method as developed in Mues et al. (2017) to estimate the CO fluxes
based on the observed CO mixing ratio and ceilometer observations of the mixing layer height
(*MLH*) in Bode for the period of 1 year (March 2013-February 2014). Using the approach used
by Mues et al. (2017), the CO fluxes can be calculated from the increase in CO concentrations
during the nighttime period when the *MLH* is nearly constant, using:

$$FCO\left(t_x, t_y\right) = \frac{\Delta\text{CO} \times ave(MLH(t_x), MLH\ (t_y))}{\Delta t \times 3600} \times \frac{MLH(t_y)}{MLH\ (t_x)} \tag{1}$$


where *FCO (t_x, t_y)* is the CO emission flux (in $\mu$g m$^{-2}$ s$^{-1}$) between time $t_x$ and $t_y$ (in hours), $\Delta$CO
is the change in CO mixing ratio (in $\mu$g m$^{-3}$) between time $t_x$ and $t_y$, *ave(MLH(t_x), MLH(t_y)* are
average of the mixing layer heights (in m) between time $t_x$ and $t_y$, *Δt* is the time interval  between
$t_x$ and $t_y$, and *MLH(t_y)/MLH(t_x)* is mixing layer collapse factor, accounting for the small change in
MLH between the night and the morning hours. The calculation of the emission flux is based on
mean diurnal cycle per month of CO and MLH and tx and ty represent the time with the
minimum (tx) and the maximum (ty) CO concentration in the night and morning (see Mues et al.,
2017 for details).

This method of calculating the CO emission flux is based on five main assumptions: (i) CO is
well-mixed horizontally and vertically within the mixing layer in the region immediately
surrounding the Bode site; (ii) the *MLH* remains fairly constant during the night so that the
product of the CO concentration ($\mu$g m$^{-3}$) and the *MLH* (m) represents CO mass per unit area
within the column, and any change in mass per unit area represents the net flux into the column;
(iii) the transport of air pollutants into and out of the stable nocturnal boundary layer of the
valley is negligible, which is supported by the calm winds (<1 m s$^{-1}$) during the night and
morning hours at the site (Mahata et al., 2017), (iv) the vertical mixing of pollutants between the
mixing layer and the free atmosphere is assumed to be negligible at night, thus strictly seen is the
estimated CO flux calculated with eq. 1 only valid for the morning hours. When applied to the
whole day the implicit assumption is that the emissions are similar during the rest of the 24 h
period.  An assumption that is viable on average for some sources like brick kilns which operate
day and night, but which does not apply to all sources, e.g., the technique will tend to
underestimate emissions due to traffic, which are typically much stronger during the day than at
night, while it will overestimate emissions due to waste burning, which is typically more
prevalent during the night and early morning (pre-sunrise) than during the daytime.  Assumption
(iv) is made because equation 1 only works well for calculating the CO flux at night-morning
period, when there is a relatively constant *MLH* and limited vertical and horizontal mixing; and
v) CO emission is assumed to be uniform throughout the valley; this may not be correct, but
cannot be verified until a high resolution emission inventory data is available, which is being
developed for the Kathmandu Valley and rest of Nepal with a 1 km x 1km spatial resolution
(Sadavarte et. al., 2018). During nighttime assumption (i) might not be entirely correct since the
degree of mixing in the nocturnal stable layer and thus the vertically mixing is drastically
reduced compared to daytime (and thus the term "mixing layer" is not entirely accurate, but we
nevertheless apply it here due to its common use with ceilometer measurements).  This adds a
degree of uncertainty to the application of ceilometer observations to compute top-down
emissions estimates, which will only be resolved once nocturnal vertical profile measurements
are also available in order to characterize the nocturnal boundary layer characteristics and the
degree to which the surface observations are representative of the mixing ratios throughout the
vertical column of the nocturnal stable layer.
It is not possible to directly compute the emission flux for a full 24-hour day using this top-down
method, since the emissions during the day could be either greater or smaller than at night, and
because the other assumptions do not hold (in particular there is considerable vertical mixing
with the free troposphere and stronger horizontal transport during the daytime). Thus the top-
down computation only provides a useful indicative value.  However, while it is also not possible
to estimate how much different the daytime emissions are, it is possible to determine an absolute
lower bound for the CO flux ($FCO_{min}$) by making the extreme assumption that the CO emissions
are non-zero only during the hours which were used in the calculation, and that they were zero
during the rest of the day (this provides a lower bound to the emissions since the daytime
emissions physically cannot be negative). This lower bound of the flux ($FCO_{min}$) is thus
calculated by scaling back the 24-hour flux to only applying over the calculation time interval
($\Delta t$), using:

$$FCO_{min.} = FCO \times \frac{\Delta t}{24} \qquad (2)$$


Figure 8 shows the estimated monthly CO emission flux, along with its 25[th] and 75[th] percentile
values as an indication of the variability of the estimated flux in each month; the lower bound of
the CO flux based on Equation 2 is also shown. The estimated annual mean CO flux at Bode is
4.9 µg m$^{-2}$ s$^{-1}$. Seasonally, the emissions are computed to be highest during December to April
(3.6-8.4 µg m$^{-2}$ s$^{-1}$), coinciding with the brick kiln operation period, which resulted in elevated
concentrations of most pollutants at Bode (Kim et al., 2015; Chen et al., 2016; Sarkar et al.,
2016; Mahata et al., 2017; Mues et al., 2017), including CO (Bhardwaj et al., 2017; Mahata et
al., 2017), while the emissions were generally lower during the remaining months (0.5-5.4 µg m$^{-}$
$^{2}$ s$^{-1}$). The uncertainty in the top-down CO emissions estimate will be largest during June to
October, due to the greater diurnal and day-to-day variability with the minimum and maximum
CO mixing ratio values during the night and early morning used in Equation 1 often being less
distinct than in the other months.

Comparing the annual mean top-down estimated CO emission flux at Bode (4.9 µg m$^{-2}$ s$^{-1}$) with
available global and regional emission inventories, the top-down estimated CO flux is twice the
value, 2.4 µg m$^{-2}$ s$^{-1}$, for the Kathmandu Valley in the EDGAR HTAP V2.2 emission inventory
database for 2010 [note that the CO emission values for the location at Bode and the whole
averaged for the valley (27.65-27.75°N, 85.25-85.40°E) were found to be the same up to two
significant figures]. The estimated CO flux was 6.5-8 times as high as in the REAS database
(0.63-0.76 µg m$^{-2}$ s$^{-1}$, based on the 2008 values in Kurokawa et al., 2013), and between 3 and 14
times higher than the values in the INTEX-B database for 2006 (0.35-1.77 µg m$^{-2}$ s$^{-1}$; Zhang et
al., 2009). The large differences between our estimated CO emission flux and these emission
databases is not likely to be due to the comparison of data for different years; rather, it indicates
the substantial uncertainties in both the top-down and bottom-up approaches to estimating the
emission flux. Although our approximation of the emission flux relies on several assumptions,
the fact that the lower bound value that we calculate is still as high as or higher than the values in
some of the published emission datasets likely indicates that the bottom-up emissions are
missing or underestimating some important sources, which will be important to examine
carefully and improve as a basis for interpreting future modelling studies of CO pollution in the
Kathmandu Valley and surrounding regions, as well as for assessing possible mitigation options.

The emission estimates computed here are subject to several further uncertainties which are
discussed in detail in Mues et al., (2017). In short, the uncertainties of CO flux estimates arise
from (i) the assumptions that Bode site represents the whole atmospheric column and entire
valley, which is not possible to verify without having many simultaneous monitoring stations in
the valley (measurements at a few sites where CO was monitored for this study show some
difference in CO mixing ratios), (ii) the higher variability (unclear minima and maxima during
the morning and night hours) in the diurnal cycles of CO from June to October show a much
higher variability than other months, that in turn makes it difficult to choose the exact hour of
CO minimum and maximum needed for the flux estimation and (iii) the possible impact of wet
deposition is not taken into account but would rather cause to generally underestimate the
emission rate.

## 4. Conclusions

Ambient CO and $O_3$ mixing ratios were measured in the framework of the SusKat-ABC
international air pollution measurement campaign at five sites (Bode, Paknajol, Bhimdhunga,
Naikhandi and Nagarkot) in the Kathmandu Valley (Table 1) and its fringes, initially during
January to July 2013, and later extended to one year at three sites (Bode, Paknajol and Nagarkot)
to better understand their seasonal characteristics. The observed CO and $O_3$ levels at all sites
except Nagarkot were characteristic of highly-polluted urban settings, with the particular feature
that the bowl-shaped valley and resulting meteorology had several effects on the pollution levels.

At all sites, the CO mixing ratios were higher during the early morning and late evening, especially an observation connected to the interplay between the ventilation of the boundary layer and the diurnal cycles of the emission sources. Under calm wind conditions that limited mixing within, into and out of the Kathmandu Valley, the morning CO peak tended to be more pronounced due to the buildup of pollution at night in the shallow planetary boundary layer. This nocturnal buildup was especially strong during January to April at Bode, with the mean CO mixing ratio increasing by about a factor of 4 in the 12 hours from 20:00 to 08:00, especially due to operation of nearby brick kilns continuing through night. During the daytime, the wind becomes stronger and the horizontal and vertical circulation dilutes and transports pollution around and out of the valley. Although normally the pollution levels are presumed to be higher in the heavily populated valley than in the immediate surrounding region, occasionally the synoptic circulation will transport in CO and $O_3$-rich air, especially influenced by forest fires and agro-residue burning in the IGP region and Himalayan foothills, as was observed on a few episodes in the pre-monsoon season.

The observed $O_3$ mixing ratio was highest in the pre-monsoon season at all sites, and the daily maximum 8-hour average $O_3$ exceeded the WHO guideline of 50 ppb on about 80% of the days during this season at the semi-urban/urban sites of Bode and Paknajol, while at Nagarkot (which is in the free troposphere, i.e., above valley's boundary layer most of the time, especially during nighttime) it exceeded the WHO guideline on 92% of the days in pre-monsoon season. During the whole observation period, the 8 hour maximum average $O_3$ exceeded the WHO recommended value on 29%, 37% and 45% of the days at Bode, Paknajol and Nagarkot, respectively. The diurnal cycle showed evidence of photochemical production, larger scale advection of polluted air masses as well as possible down-mixing of $O_3$ during the daytime, as also observed by Putero et al., (2015) at Paknajol, with the hourly mixing ratio at the polluted site increasing from typically 5-20 ppb in the morning to an early afternoon peak of 60-120 ppb (Putero et al., 2015; Bhardwaj et al., 2018).

These high $O_3$ levels have deleterious effects on human health and ecosystems, including agro-ecosystems in the Kathmandu Valley and surrounding regions, thus justifying mitigation measures to help reduce the levels of $O_3$ (its precursors VOCs and NOx), CO and other pollutants. Determining the most effective mitigation measures will be challenging due to the

complicated interplay of pollution and meteorology as well as local and regional pollution sources. This study has provided information on current ambient levels and the diurnal/seasonal variations. This will be helpful in the design of future policies, both as a baseline for evaluating the effectiveness of mitigation measures, as well as giving insight into the connections between various pollutant sources (e.g., brick kilns) and their impacts on seasonally elevated CO levels, especially at nighttime. One particular contribution has been the development of a top-down estimate of the total emission flux of CO at Bode, which was found to be 4.9 µg m$^{-2}$ s$^{-1}$. This is several times higher (by a factor of 2-14 times) than the CO emission fluxes for the Kathmandu Valley in state-of-the-art inventories such as EDGAR-HTAP, REAS, and INTEX-B. This points out the need for the development of updated comprehensive emission inventory databases for this region. The improved emission inventory is necessary to provide more accurate input to model simulations to assess air pollution processes and mitigation options for the Kathmandu Valley and the broader surrounding region.

While the high levels of particulate pollution in the Kathmandu Valley have caught the main attention of the public and policymakers, due to their immediately visible nature, our paper points out that ozone is also a serious problem here. In fact, its higher levels on the nearby mountaintop location of Nagarkot, which is much more representative of regional air pollution, point to an ozone problem in the wider foothills of the Himalayas. In fact the extent of ozone pollution in the large surrounding Himalayan foothills has been insufficiently recognized until our study. This needs monitoring and research to identify feasible mitigation options.

**Data Availability**

The observational data collected for this study will be made public through the SusKat website of IASS. They are also available upon direct request sent to maheswar.rupakheti@iass-potsdam.de and khadak.mahata@iass-potsdam.de.

**Acknowledgement**

We are thankful to the funders of the IASS – the German Ministry of Education and Research (BMBF) and the Brandenburg State Ministry of Science, Research and Culture (MWFK) – for

their generous support in making these measurements and their analysis possible. This study was
partially supported by core funds of ICIMOD contributed by the governments of Afghanistan,
Australia, Austria, Bangladesh, Bhutan, China, India, Myanmar, Nepal, Norway, Pakistan,
Switzerland, and the United Kingdom, as well as by funds from the Government of Sweden to
ICIMOD's Atmosphere Initiative. The authors would like to thank Bhupesh Adhikary,
Bhogendra Kathayat, Shyam Newar, Dipesh Rupakheti, Nirjala Koirala, Ashish Bhatta, Begam
Roka, Sunil Babu Khatry, Giampietro Verza, and several staff members at the Kamdhenu
Madhyamik Vidhyalaya, Naikhandi who assisted in the field measurements, Siva Praveen
Puppala for initial data processing, and Pankaj Sadavarte for helping with the emission
databases. We are grateful to the Department of Environmental Sciences, University of Virginia,
USA, for making available CO and $O_3$ instruments for the measurements. We also thank the staff
at Real Time Solutions (RTS), Lalitpur, Nepal for providing an automatic weather station.

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

**Table 1.** Information on the sampling sites (of the SusKat-ABC campaign) used in this study with sampling carried out during 2013-2014 in the Kathmandu Valley. The altitude is in meter above mean sea level (m asl)

| Site | General setting of site | Location, altitude (m asl) |
|---|---|---|
| Bode | Sub-urban, tallest building with scattered houses surrounded by agricultural fields | 27.69°N,  85.40°E,  1345 |
| Bhimdhunga | Rural. On the ridge, close to the pass separating the Kathmandu Valley from a valley of a tributary the Trishuli River to the west | 27.73°N, 85.23°E,  1522 |
| Paknajol | Urban, city-center, the tallest building in the neighborhood | 27.72°N, 85.30°E, 1380 |
| Naikhandi | Rural, at outlet of Bagmati River in Southwest corner of the Valley | 27.60°N, 85.29°E, 1233 |
| Nagarkot | Mountain rural. Mountain top site of the eastern valley rim, north facing towards the Kathmandu Valley | 27.72°N , 85.52°E, 1901 |

**Table 2.** Details of the instruments deployed at different sites during the observation period during January 2013-March 2014 in the Kathmandu Valley.

| Location | Instrument | Parameters | Inlet/sensor height (above ground) | Duration | Group |
|---|---|---|---|---|---|
| 1. Bode | a. Horiba APMA-370 | CO | 20 m | 1 Jan-7 Jun 2013 | ARIES |
| | b. Teledyne 400E | $O_3$ | 20 m | 1 Jan-7 Jun 2013 | ARIES |
| | c. Thermo Scientific 49i | $O_3$ | 20 m | 18 Jun-31 Dec 2013 | IASS |
| | d. Picarro G2401 | CO | 20 m | 6 Mar 2013-5 Mar 2014 | ICIMOD |
| | e. Campbell AWS | T, RH, SR, WS, WD, RF | 22 m | 1 Jan-30 Mar 2013 | IASS |
| | f. Davis AWS (Vantage Pro2) | T, RH, P, RF | 21 m | 30 May-Jul 2013 | UVA |
| | g. Ceilometer (Vaisala CL31) | MLH | 20 m | 01 Mar 2013- 28 Feb 2014 | JGUM |
| 2. Bhimdhunga | a. Thermo Scientific 48i | CO | 2 m | 1 Jan-15 Jul 2013 | UVA |
| | b. AWS Hobo Onset | T, RH, SR, WS, WD, P | 5 m | 1 Jan-30 Jun 2013 | UVA |
| 3. Naikhandi | a. Thermo Scientific 48i | CO | 5 m | 3 Jan- 6 Jun 2013 | UVA |
| | b. 2B Tech. Model 205 | $O_3$ | 5 m | 1 Feb-25 May 2013 | UVA |
| | c. AWS Hobo Onset | T, RH, SR, WS, WD, P | 2 m | 3 Jan-25 Apr 2013 | UVA |
| 4. Nagarkot | a. Thermo Scientific 48i | CO | 5 m | 13 Feb-Apr 3 2013; 8 Jun-15 Jul 2013 | UVA |
| | b. Thermo Scientific 49i | $O_3$ | 5 m | 9 Jan-30 Jun 2013 | UVA |
| | c. Campbell AWS | T, RH, SR, WS, WD, RF | 7 m | | IASS |
| | d. AWS (Vaisala WXT 520) | T, RH, SR, WS, WD, RF, P | 7 m | 10 Feb-30 Jun 2013 | RTS |
| 5. Paknajol | a. Thermo Environmental (49i) | $O_3$ | 25 m | 1 Feb 2013-30 Jan 2014 | EV-K2-CNR |
| | b. AWS (Vaisala WXT 425) | T, RH, SR, WS, WD, RF, P | 25 m | 1 Feb 2013-30 Jan 2014 | EV-K2-CNR |

Note: T - temperature, RH - relative humidity, SR- solar radiation, WS - wind speed, WD - wind direction, RF- rainfall, P – pressure and MLH – Mixing layer height; ARIES - Aryabhatta Research Institute of Observational Sciences, India; ICIMOD - International Center for Integrated Mountain Development, Nepal; IASS - Institute for Advanced Sustainability Studies, Germany; UVA- University of Virginia, USA; JGUM – Johannes Gutenberg University Mainz, Germany; RTS - Real Time Solutions, Nepal; Ev-K2-CNR - Everest-Karakorum - Italian National Research Council, Italy.

**Table 3.** Summary of the monthly average ozone mixing ratios (ppb) [average (Avg), standard deviation (SD), minimum (Min.) and maximum (Max.)] at four sites* in the Kathmandu Valley, Nepal during 2013-2014 and two sites (Manora Peak and Delhi) in India

| Month | Bode Avg ± SD [Min., Max.] | Paknajol Avg ± SD [Min., Max.] | Nagarkot Avg ± SD [Min., Max.] | Manora[a] Peak Avg ± SD | Delhi[b] Avg [Min., Max.] |
|---|---|---|---|---|---|
| January | 23.5 ± 19.9 [1.4, 87.1] | 16.9 ± 18.3 [0.1, 71.7][*] | 46.7 ± 5.7 [36.4, 73.7 | 37.3 ± 14.8 | 19.3 [10, 14.7] |
| February | 25.6 ± 20.4 [1.2, 94.5] | 24.2± 20.1 [1.6, 91.7] | 47.5 ± 7.5 [28.2, 83.6] | 43.8 ± 16.8 | 25.3 [10.9, 55.7] |
| March | 37.4 ± 24.3 [1.2, 105.9] | 37.7 ± 23.8 [1.6, 95.8] | 62.4 ± 9.5 [40.5, 98.9] | 56.6 ± 11.4 | 29.7 [13.8, 58] |
| April | 43.4 ± 26.6 [1.4, 116.2] | 46.7 ± 26.8 [1.0, 115.5] | 71.5 ± 15.5 [40.1, 121.0] | 63.1 ± 11.7 | 33 [13.7, 64.3] |
| May | 38.5 ± 21.2 [2.0, 111.1] | 42.8 ± 20.6 [6.7, 103.3] | 59.0 ± 20.6 [15,0, 124.5] | 67.2 ± 14.2 | 35.4 [19.8, 62] |
| June | 27.8 ± 12.0 [1.7, 68.4] | 27.5 ± 17.0 [0.6, 90.7] | 34.2 ± 9.1 [4.6, 72.0] | 44.0 ± 19.5 | 25.6 [12.8, 46.4] |
| July | 21.1 ± 9.5 [1.7, 82.0] | 20.5 ± 13.4 [2.0, 77.9] | 25.9 ± 6.2 [11.1, 48.0] | 30.3 ± 9.9 | 19.1 [9.4, 37.1] |
| August | 20.3 ± 9.9 [2.0, 70.9] | 20.1 ± 12.6 [0.8, 73.1] | 28.3 ± 5.8 [15.5, 62.9] | 24.9 ± 8.4 | 14.3 [9.7, 29.5] |
| September | 23.3 ± 14.9 [0.5, 85.9] | 24.9 ± 17.4 [0.4, 108.1] | 34.8 ± 9.6 [16.1, 79.7] | 32.0 ± 9.1 | 17.7 [7.7, 37.7] |
| October | 19.4 ± 13.8 [0.1, 70.9] | 22.6 ± 17.0 [0.6, 83.5] | 35.2 ± 10.2 [18.0, 73.8] | 42.4 ± 7.9 | 21.7 [9, 56.9] |
| November | 18.6 ± 15.1 [0.3, 67.7] | 22.4 ± 20.9 [0.1, 84.0] | 40.1 ± 8.1 [25.6, 73.3] | 43.9 ± 7.6 | 22.6 [9, 55.1] |
| December | 21.7 ± 17.8 [1.0, 96.6] | 19.5 ± 19.7 [0.1, 82.0] | 43.8 ± 9.0 [24.8, 85.11] | 41.6 ± 6.3 | 20.2 [9.1, 40.3] |
| Season: | | | | | |
| Winter | 24.5 ± 20.1 [1.2, 94.5] | 20.2 ± 19.6 [0.1, 91.7] | 45.8 ± 7.8 [24.8, 85.1] | 40.9 | 21.6 [9.1, 55.7] |
| Pre-monsoon | 39.8 ± 24.2 [1.2, 116.2] | 42.4 ± 24.0 [1.0, 115.5] | 64.3 ± 16.7 [14,9, 124.5] | 62.3 | 32.7 [13.7, 64.3] |
| Monsoon | 22.7 ± 12.0 [0.5, 85.9] | 23.2 ± 15.5 [0.4, 108.1] | 30.8 ± 8.7 [4.6, 79.7] | 32.8 | 19.2 [7.7, 46.4] |
| Post-monsoon | 19.0 ± 14.5 [0.1, 70.9] | 22.5 ± 18.9 [0.1, 84.0] | 37.6 ± 9.5 [18.0, 73.8] | 39.4 | 22.2 [9, 56.9] |

[a] Kumar et al. (2010), [b] Ghude et al. (2008). * $O_3$ data of Paknajol on January was of 2014.

**Table 4.** Averege CO mixing ratio (ppb) at different time of the day (daytime - 12:00 – 16:00), and nighttime - 23:00 – 03:00) and the monthly average (total) at four sites in the Kathmandu Valley.

| Sites | Winter (16 Jan-15 Feb) | | | Pre-monsoon (16 Mar-15 Apr) | | | Monsoon (16 Jun-15Jul) | | | Post-monsoon (16 Oct-15 Nov) | | |
|---|---|---|---|---|---|---|---|---|---|---|---|---|
| | daytime | nighttime | Total | daytime | nighttime | total | Daytime | nighttime | total | daytime | nighttime | total |
| Bode | 405.35 | 927.21 | 819.17 | 430.91 | 839.17 | 770.52 | 210.59 | 230.08 | 241.34 | 269.10 | 453.95 | 397.24 |
| Bhimdhunga | 324.62 | 354.23 | 374.27 | 374.64 | 479.37 | 471.33 | 196.61 | 202.85 | 198.40 | | | |
| Naikhandi | 280.97 | 356.14 | 380.40 | 382.71 | 425.17 | 449.83 | | | | | | |
| Nagarkot | | | | | | | 141.68 | 158.78 | 160.41 | | | |

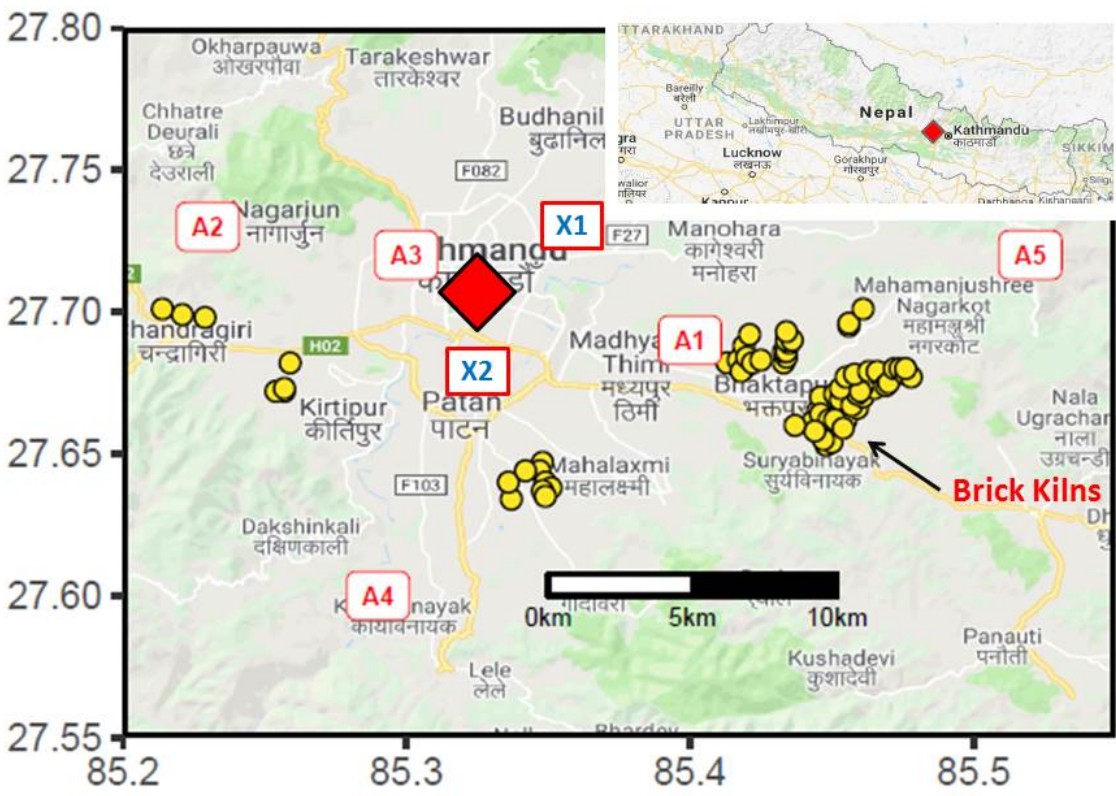

**Figure 1.** Observation sites in the SusKat-ABC international air pollution campaign during 2013-2014 in the Kathmandu Valley. A1 = Bode, A3 = Paknajol, and A4 = Naikhandi were selected within the valley floor and A2 = Bhimdhunga and A5 = Nagarkot on the mountain ridge. Naikhandi site is also near the Bagmati River outlet. Past study sites, Bouddha (X1) and Pulchowk (X2), which are referred in the manuscript, are also shown in the Figure. Source: Google Maps.

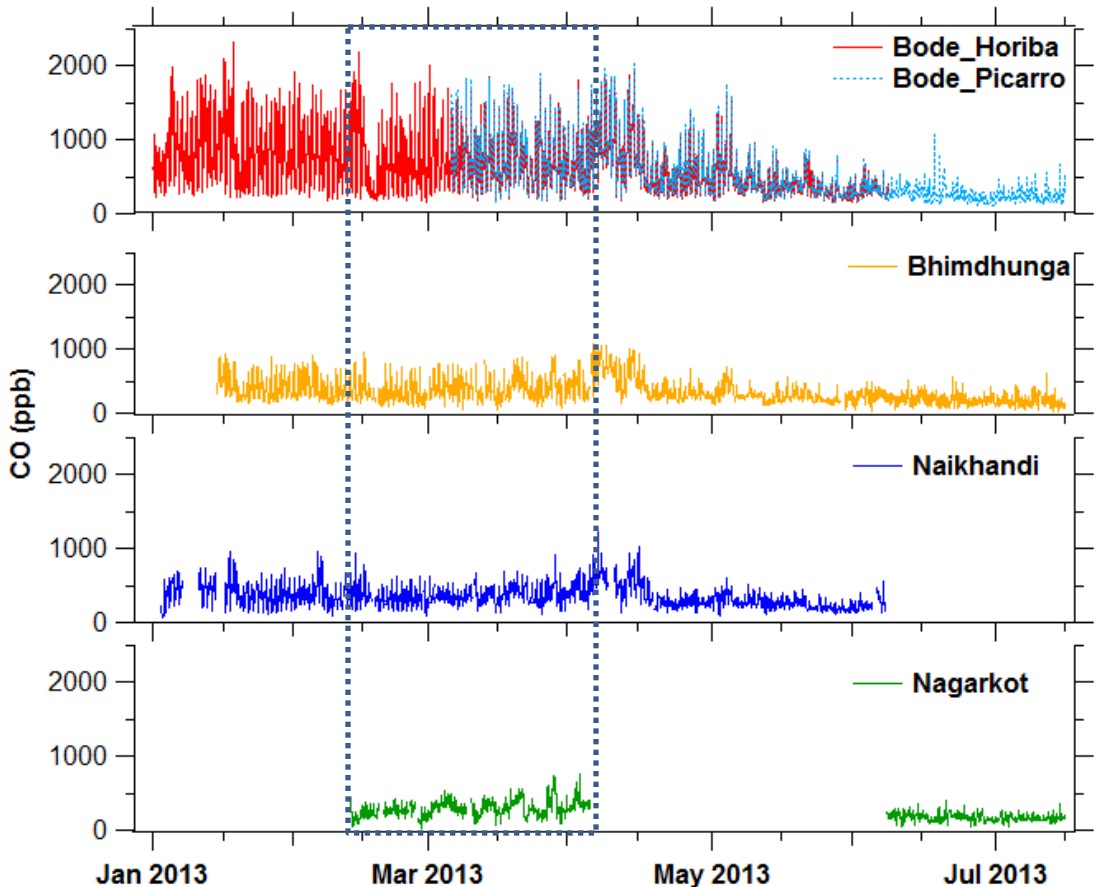

**Figure 2.** Hourly average CO mixing ratios observed at supersite (Bode) and three satellite sites (Bhimdhunga, Naikhandi and Nagarkot) of the SusKat-ABC international air pollution measurement campaign during January to July 2013 in the Kathmandu Valley. The dotted box represents a period (13 February - 03 April, 2013) during which data for all four sites were available.

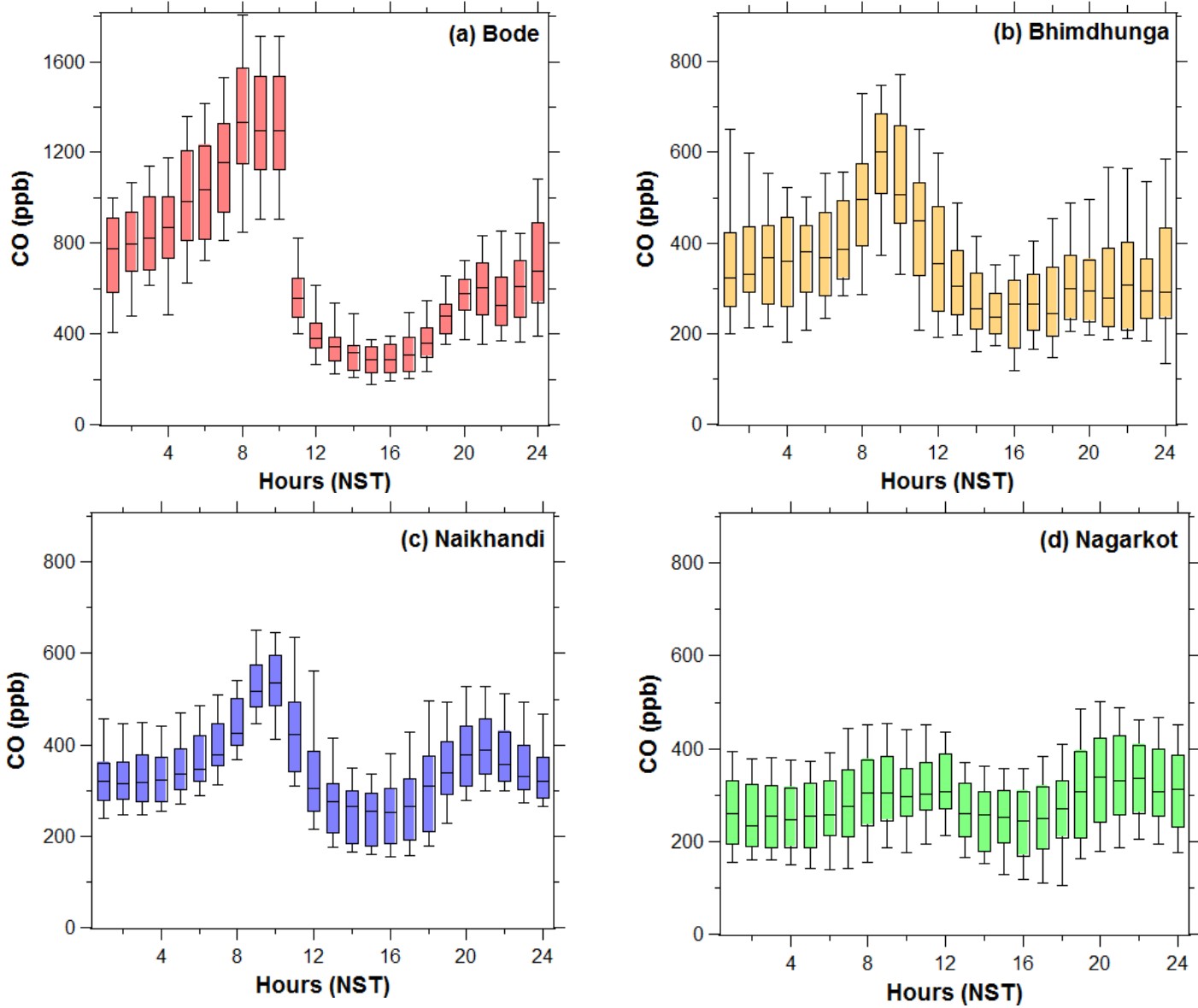

**Figure 3.** Diurnal variations of hourly average CO mixing ratios during the common observation period (13 February–03 April, 2013) at Bode, Bhimdhunga, Naikhandi and Nagarkot. The lower end and upper end of the whisker represents 10[th] and 90[th] percentile, respectively; the lower end and upper end of each box represents the 25[th] and 75[th] percentile, respectively, and the black horizontal line in the middle of each box is the median for each month. Note: the y-axis scale of Bode is twice that of the other three sites.

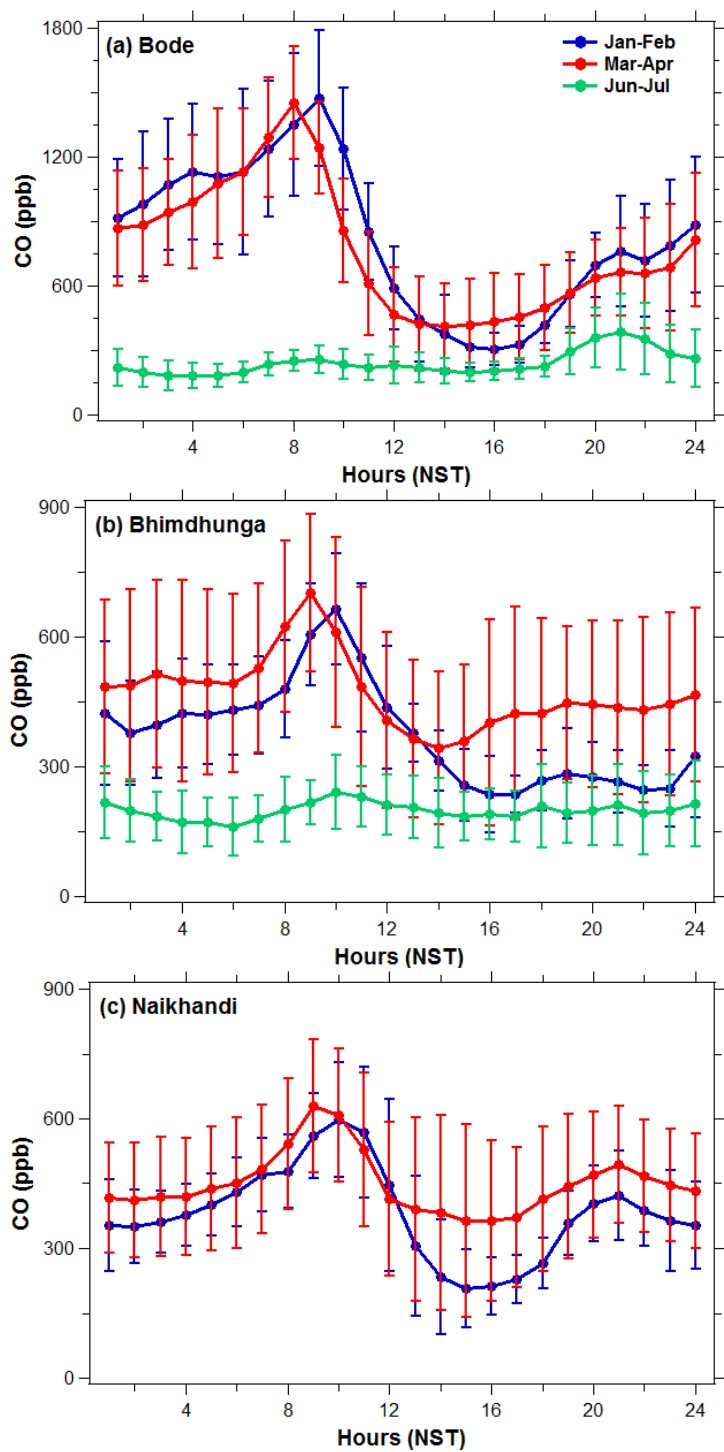

**Figure 4.** Comparison of diurnal variation of hourly average CO mixing ratios for four seasons at Bode,Bhimdhunga and Naikhandi. Due to the lack of continuous data at some sites, data of one month in each season were taken for comparison as representative of the winter (16 Jan – 15 Feb), pre-monsoon (16 Mar – 15 Apr) and monsoon (16 Jun – 15 Jul) season of 2013. Note: y-axis scale of the top panel (Bode) is double than lower two panels (Bhimdhunga and Naikhandi).

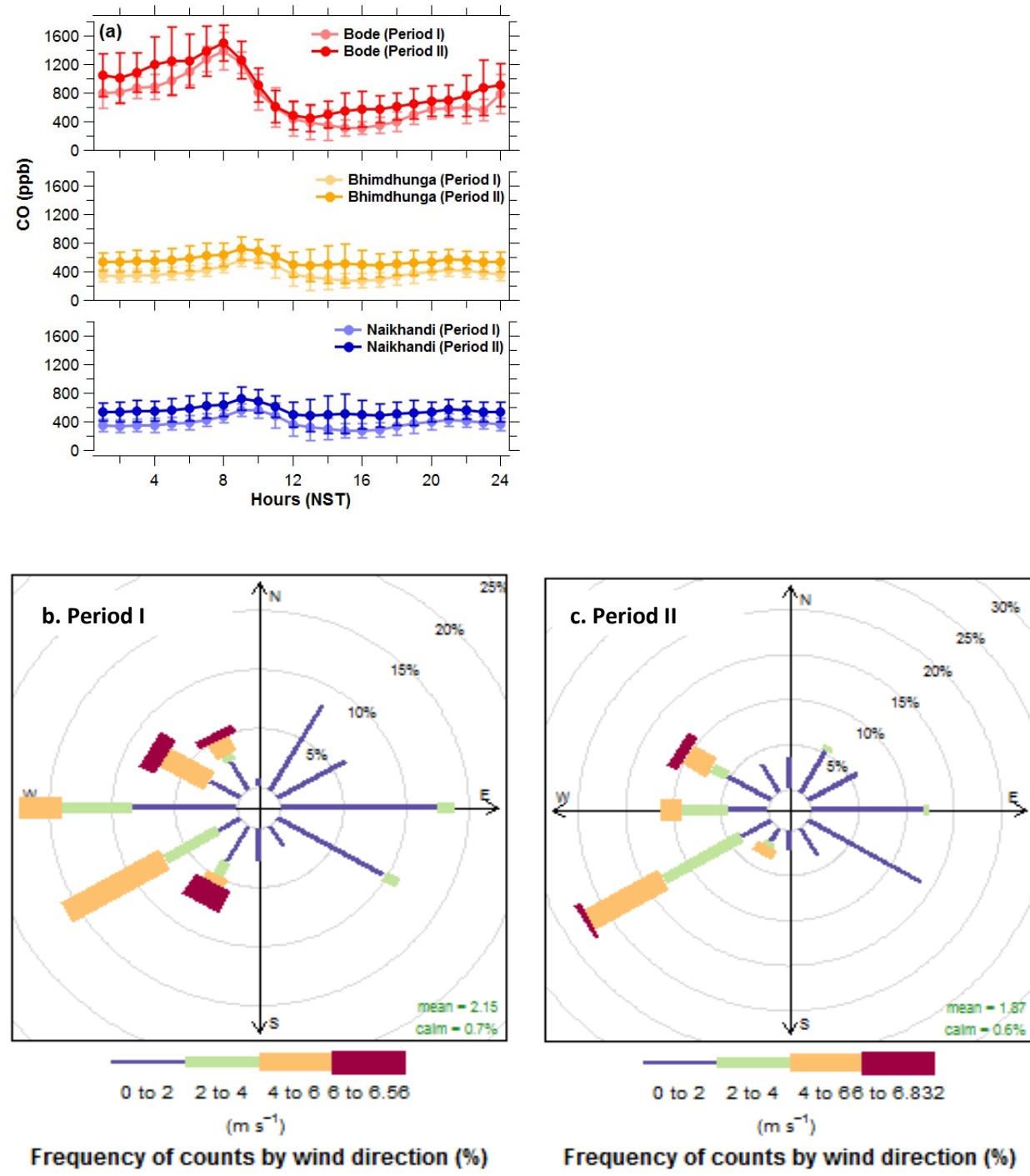

**Figure 5.** Comparison of hourly average CO mixing ratios during normal days (March 16-30), labelled as period I (faint color) and episode days (April 1-15), labelled as period II (dark color) in 2013 at (a) Bode, Bhimdhunga and Naikhandi in the Kathmandu Valley. The wind roses at Bode corresponding to two periods are also plotted (b) period I and (c) period II respectively.

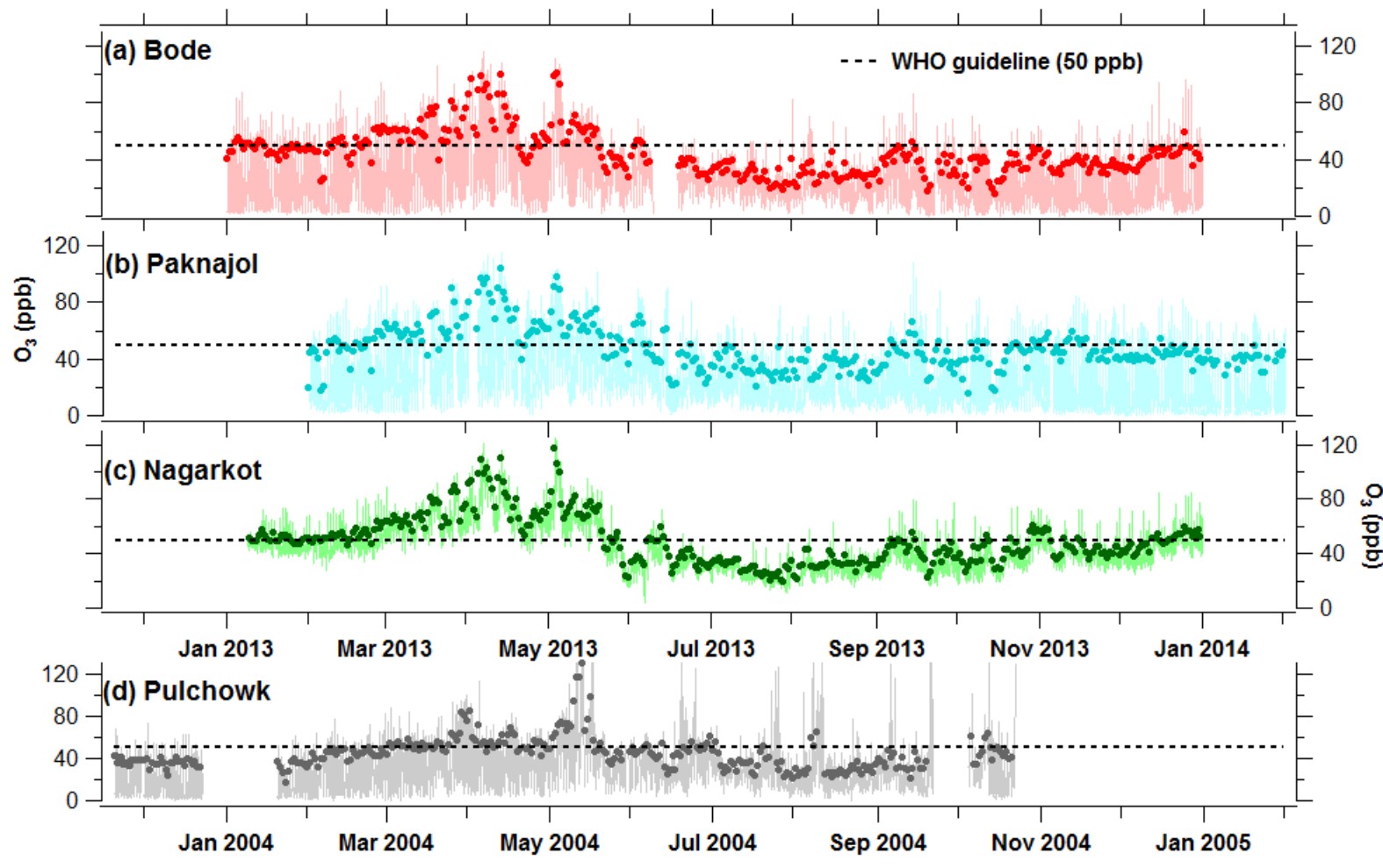

**Figure 6.** Time series of hourly average (faint colored line) and daily maximum 8-hr average (solid colored circle) $O_3$ mixing ratio at (a) Bode (semi-urban), (b) Paknajol (urban)and (c) Nagarkot (hilltop) observed during 2013-2014, and (d) Pulchowk (urban) observed during November 2003-October 2004 in the Kathmandu Valley. Black dotted line represents WHO guideline (50 ppb) for daily maximum 8-hour average of $O_3$.

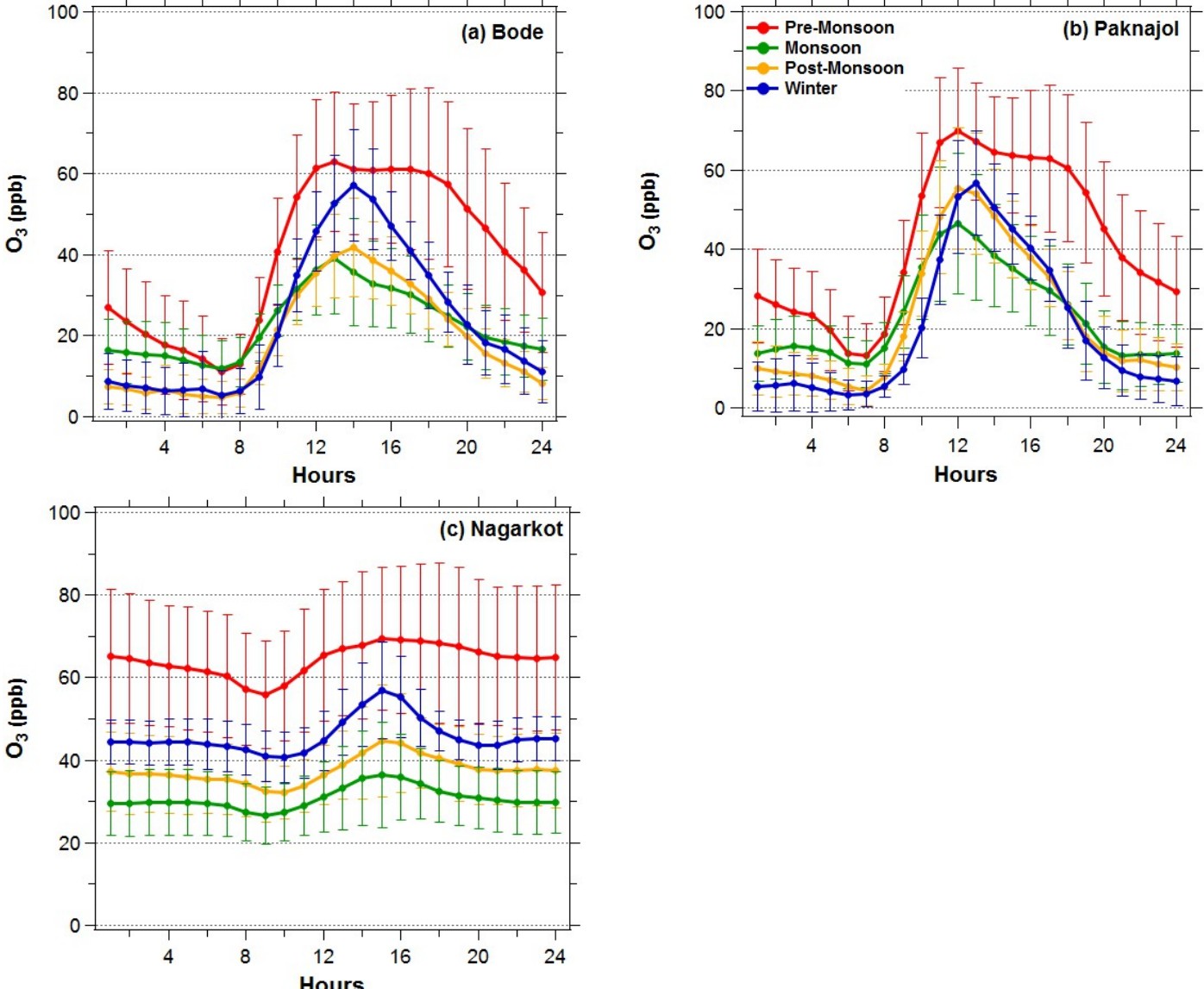

**Figure 7.** Diurnal pattern of hourly average $O_3$ mixing ratio for different seasons during January 2013-January 2014 at (a) Bode, (b) Paknajol, and (c) Nagarkot in the Kathmandu Valley. The four seasons (described in the text) are defined as: pre-monsoon (Mar-May), monsoon (Jun-Sep), post-monsoon (Oct-Nov), winter (Dec-Feb).

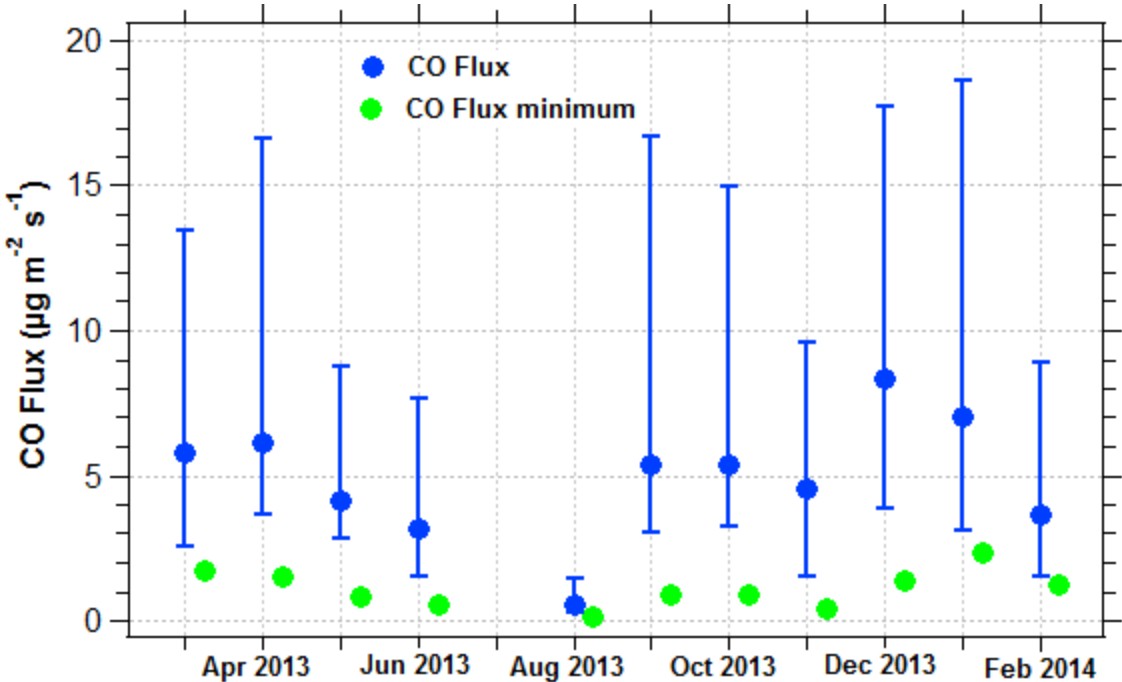

**Figure 8.** The estimated monthly average CO emission flux, which is based on the mean diurnal cycle of CO mixing ratios of each month for two conditions: (i) with data of all days (CO Flux) (blue dot) with lower and upper ends of the bar representing 25[th] and 75[th] percentile respectively, and (ii) with data of morning hours (CO Flux minimum (green dot) in which zero emission is assumed for the other hours of the day. The fluxes for July were not estimated as there were insufficient (less than 15 days) of concurrent CO and mixing layer height data. It is expected that the $F_{CO}$ and $F_{COmin.}$ for July should fall between values for June and August 2013.