# Peer review of "Observation and analysis of spatio-temporal characteristics of surface ozone and carbon"

_Atmospheric Chemistry and Physics, 2017_

## Referee Comment (RC1) · Anonymous Referee #1 · 10 Jan 2018

**General remarks**

This paper reports air pollution (ozone and CO) in the Kathmandu area over longer time periods than hitherto available. Air pollution in this region is an important problem and reliable information covering all seasons is an important contribution to research on these issues. And I agree with the authors that the high ozone mixing ratios observed during the pre-monsoon period is of a high concern for human health and ecosystems, in the region. Here I would encourage the authors to go beyond what is presented in

the paper and (briefly) discuss possible mitigation options (following the idea of "policy relevant, not policy prescriptive").

However, I have also some reservations about the interpretation of some aspects of the reported data and also some issues with the presentation. I suggest taking these points into account when revising the paper. If this is done in an appropriate way, I suggest that the Editor accepts the paper for publication in ACP.

**Comments in detail**

One aspect that is only discussed in passing in the paper is the role of stratospheric intrusions as a source of ozone in the upper troposphere in the region (e.g., Wang et al., 2012). Thus, ozone at higher altitudes in the troposphere could be enhanced independent of tropospheric pollution. I suggest that this aspect should be better discussed in the paper.

Further, I suggest more comparison of the ozone pollution found at the Kathmandu valley with pollution levels elsewhere in the world (e.g. Huszar et al., 2016) . Are the close to zero ozone values reported here (due to NO titration) also found in other regions of the world? These question is important for mitigation strategies, because to achieve significant ozone reduction over cities in central Europe, the emission control strategies have to focus on the reduction of VOCs (Huszar et al., 2016).

I repeat my comment on Fig. 1 from the initial/quick review here: I find the Google Earth figure not appropriate. The yellow pins are strange and the blue letters are difficult to read against the background. I suggest changing to a figure showing the locations of the sites in a map showing the orography clearly.

I also suggest to state the calender months, not just the seasons. This is done in l. 271, but it should also be stated in the introduction and in the abstract.

The value of for the CO flux at Bode is given to three significant numbers, is this really appropriate? Do you have an error estimate for this number? I think this value is an important result from this study so it deserves some attention.

Finally, I could very well imagine that the data presented in this paper are of interest to other researchers as well. Therefore I suggest to add a comment on data availability to the paper.

**Minor issues**

- l. 31: drop 'on'

- l. 32: 'pollutants'

- l. 37: add altitude for Naikhandi

- l. 42: State 'how long' extended

- l. 46: state the calender months, not everybody is familiar with these seasons.

- l. 46/47: 'due to the emissions from brick kiln industries' How do you know? How much of this is speculation/hypothesis how much is really shown in the paper?

- l.50: in which way did the meteorology play a role?

- l. 52: 'Some influence' is a bit vague, can you be more specific here?

- l. 54: The value of 4.92 is given to three significant numbers, is this really appropriate? Do you have an error estimate for this number?

- l. 63: 'as well as': which effect dominates?

- l. 65: on the basis of which assessment can you say 'due to'?

- l. 80: one further impact of local pollution could also be convective uplift to tropopause altitudes and transport into the extra-topical stratosphere in the monsoon season (e.g. Tissier and Legras, 2016, and references therein).

- l. 93: 2017 → 2018

- l. 97: also toxic outdoors?

- l. 115: measured → reported measurements

- l. 133: for the Kathmandu . . .

- l. 167: $O_3$

- l. 227: define 'AWS'

- l. 286: due to a problem

- l. 320, 321: How do you know?

- l. 339: CO mixing ratios

- l. 453: stratospheric intrusions are mentioned here but only in passing.

- l. 472: make → draw

- l. 506: give altitude of Nagarkot here. Also the statement here is a bit vague, can you be more quantitative here (instead of 'but is also').

- Mues et al. (2017); citation is missing

- l. 546: be specific what is meant with 'this'
- l. 611: change to 'an observation connected to'

- l. 617: drop 'the'

- l. 622: episode days → episodes

- l. 632: are these ozone values typical for down-mixing?

- l. 646-650: perhaps two sentences here

- l. 711: This paper is now accepted

- Figs. 5 and 7: can you show error bars in these figures?

**References**

Huszar, P., Belda, M., and Halenka, T.: On the long-term impact of emissions from central European cities on regional air quality, Atmos. Chem. Phys., 16, 1331–1352, doi:10.5194/acp-16-1331-2016, 2016.

Tissier, A.-S. and Legras, B.: Convective sources of trajectories traversing the tropical tropopause layer, Atmos. Chem. Phys., 16, 3383–3398, doi:10.5194/acp-16-3383-2016, 2016.

Wang, Y., Konopka, P., Liu, Y., Chen, H., Müller, R., Plöger, F., Riese, M., Cai, Z., and Lü, D.: Tropospheric ozone trend over Beijing from 2002–2010: Ozonesonde measurements and modeling analysis, Atmos. Chem. Phys., 12, 8389–8399, doi:10.5194/acp-12-8389-2012, 2012.

---

## Referee Comment (RC2) · Anonymous Referee #2 · 2 Mar 2018

The manuscript "Observation and analysis of spatio-temporal characteristics of surface ozone and carbon monoxide at multiple sites in the Kathmandu Valley, Nepal" by Mahata and co-authors provides an analysis of CO and $O_3$ measurements carried out at 4 sites in the Kathmandu valley during the course of one year. Due to this good data coverage the analysis allows for a more thorough analysis than previously possible and also provides some valuable CO emission estimates. The paper is well written and organised and after minor corrections suitable for publication in ACP.

**Minor comments**

[Figure]

L46 and elsewhere: Here a strong statement is made about the significant contribution of brick kilns to the observed CO concentrations. However, there is little actual proof of this shown in the manuscript. This could be improved by indicating the location of the kilns in relation to the measurement locations and a more thorough analysis/description of the nighttime wind pattern. Both of which would allow for a more creditable source attribution. Since there were also other atmospheric tracers measured at Bode, couldn't one of them (e.g. $SO_2$ also be used to support the kiln contribution?

L50,51: Please mention in which way meteorology played a key role.

L72: Please split this number into casualties due to indoor and outdoor pollution. The first number seems to be the more important one in the light of your study.

L83ff: Please also mention the special topographical and meteorological conditions (poor ventilation) that characterize the basin and further deteriorate air quality.

L96: "CO is a useful tracer of urban air pollution". In the light of large contributions to CO from forest fires and agricultural waste burning (discussed later in the text), you should mention this important source as well.

L122: How does the CO emission estimate by Shrestha et al. (2013) compare with your emission estimate? Please add to the discussion in Section 3.5.

L206 and for following sites: Where was the inlet mounted? What is the total height above ground of the inlet? Repeat from table 2.

L254: These IR CO analyzers usually show a strong drift with lab temperature. Did you assure that lab temperatures varied as little as possible (AC) or did you use some additional drift correction? Once daily zero checks would probably not be sufficient. Can you rule out that part of the observed diurnal cycle of CO is due to instrument errors?

L258: What was the result of the span check? Did the instrument drift since the last span check?

L283: What are the given uncertainties? Standard deviation of hourly observations? Uncertainty of the mean?

L302: Can you mention a bit more about what is know about the kind and timing of trash burning? Are these small scale fires (individual households) or larger scale (communities/neighbourhood)? Are there any regulations on this kind of waste treatment? It is mentioned elsewhere that this happens at night? Why? Seems to be a rather simple process to tackle to improve overall air quality.

L335: "support turbulent vertical diffusion". Although this statement is absolutely true, this is already reflected by the deeper mixing layer during daytime. I suggest to reformulate in such a way that the reasons for a deeper mixing layer are given in the first sentence (heating of surface by incoming solar radiation and (secondary) higher horizontal wind speeds and turbulence production). Then only mention the flushing effect of the increased wind speeds in the second sentence. In the end, the increased horizontal wind speeds are caused by the growing mixing layer height as well, so buoyancy production of turbulence is the real cause for the increased ventilation of the surface layer, but the above discussions seems to be sufficient.

Section 3.2.1: You could also comment on the distinctly different shapes of the nighttime increase at Bode and Bhimdhunga. Bode shows an almost linear increase, which may indicate continued emissions into the local stable boundary layer, whereas Bhimdhunga shows a more isolated peak during the morning transition phase. So it would indicate that slope winds bring part of the polluted valley boundary layer up to the pass even in the early morning, which seems well possible considering the east facing slope above which the site is located. The same influence can be seen in $O_3$ at the site.

L351f: The argument about decreased forest fires and agricultural waste burning should be clarified a bit. Up to this point in the manuscript one had the impression that most of the CO at Bode was due to the brick kilns. But now the big difference between the seasons is explained through the absence of forest fires, etc and the brick

kilns are only mentioned at the very end. When and why do they actually stop production? Due to the precipitation in the monsoon season?

L360: Why is this apparent? Even if the kilns operate at night you should show that there is a direct link to the site in terms of advection direction? Isn't residential heating the more likely candidate?

L368: There is also a distinct shift in the morning peak visible at all 3 sites for the different seasons. Can you please comment on this? Probably it is just due to an earlier onset of the morning transition in Mar-Apr, but maybe changes in local emissions may play a role as well.

L393 "data means": Not clear which parameter is referred to here. Mean CO for the whole period or at a specific time of day?

L473: The comparability of the old time series with the recent may also be hampered by the difference in location and sampling height as well as a general difference in instrument calibration. These points should be mentioned as well.

L499ff: Isn't the prolonged afternoon peak due to the same regional scale transport that was responsible for elevated CO? Free tropospheric contribution alone would not explain the difference between winter and pre-monsoon. Why not carry out the same kind of analysis as for CO in Figure 5.

L505: The dip in $O_3$ in the morning transition hours once more indicates the origin from the polluted stable boundary layer.

L536 and equation 1: Why give t in hours? Why not just use seconds? Would save the conversion factor in the equation and is a better SI unit anyway!

L542ff: One additional important limitation of the method is that of regional representativeness. As is said in the text, wind speeds are low so the observed CO increase at Bode may be rather localized and the emission estimate only valid for a small area and not for the whole city or valley. This is especially important when comparing the results

with those from emission inventories that average over relatively large grid cells.

L555: Was the method actually applied to every night that had sufficient CO data? Or did you filter for low wind speed, constant MLH conditions? In which case it should be mentioned for how many nights per month the estimate was possible.

L574: Can you provide a realistic uncertainty for this estimate?

L587, the statement in brackets: Statement unclear? What do you mean by "averaged for the valley as a whole"? Did you apply the method also to other sites? Or just to Bode?

L633: Again: mention the potential larger scale advection of polluted air masses (as for CO). See comment above.

Figure1: Instead of this 3D view, it would be more beneficial to have a plain 2D map with a scale indicator that would allow to identify the distances between sites. In addition, it would be a benefit to see the location of the large point sources (kilns) in such a map as well. Topography could still be included as isolines or shading. Main traffic routes would help as well.

---

## Author Comment (AC1) · 23 Aug 2018

The comment was uploaded in the form of a supplement:
https://www.atmos-chem-phys-discuss.net/acp-2017-709/acp-2017-709-AC1-supplement.pdf

[Figure]

[Figure]

**Figure 1.** Observation sites in the SusKat-ABC international air pollution campaign during 2013-2014 in the Kathmandu Valley. A1 = Bode, A3 = Paknajol, and A4 = Naikhandi were selected within the valley floor and A2 = Bhimdhunga and A5 = Nagarkot on the mountain ridge. Naikhandi site is also near the Bagmati River outlet. Past study sites, Bouddha (X1) and Pulchowk (X2), which are referred in the manuscript, are also shown in the Figure. Source: Google Earth.

**Fig. 1.**

[Figure]

**Figure 2.** Hourly average CO mixing ratios observed at supersite (Bode) and three satellite sites (Bhimdhunga, Naikhandi and Nagarkot) of the SusKat-ABC international air pollution measurement campaign during January to July 2013 in the Kathmandu Valley. The dotted box represents a period (13 February - 03 April, 2013) during which data for all four sites were available.

Fig. 2.

[Figure]

**Figure 3.** Diurnal variations of hourly average CO mixing ratios during the common observation period (13 February–03 April, 2013) at Bode, Bhimdhunga, Naikhandi and Nagarkot. The lower end and upper end of the whisker represents $10^{th}$ and $90^{th}$ percentile, respectively; the lower end and upper end of each box represents the $25^{th}$ and $75^{th}$ percentile, respectively, and the black horizontal line in the middle of each box is the median for each month. Note: the y-axis scale of Bode is twice that of the other three sites.

**Fig. 3.**

[Figure]

**Figure 4.** Comparison of diurnal variation of hourly average CO mixing ratios for four seasons at Bode,Bhimdhunga and Naikhandi. Due to the lack of continuous data at some sites, data of one month in each season were taken for comparison as representative of the winter (16 Jan – 15 Feb), pre-monsoon (16 Mar – 15 Apr) and monsoon (16 Jun – 15 Jul) season of 2013. Note: y-axis scale of the top panel (Bode) is double than lower two panels (Bhimdhunga and Naikhandi).

**Fig. 4.**

[Figure]

**Figure 5.** Comparison of hourly average CO mixing ratios during normal days (March 16-30), labelled as period I (faint color) and episode days (April 1-15), labelled as period II (dark color) in 2013 at (a) Bode, Bhimdhunga and Naikhandi in the Kathmandu Valley. The wind roses at Bode corresponding to two periods are also plotted (b) period I and (c) period II respectively.

**Fig. 5.**

[Figure]

**Figure 6.** Time series of hourly average (faint colored line) and daily maximum 8-hr average (solid colored circle) O₃ mixing ratio at (a) Bode (semi-urban), (b) Paknajol (urban)and (c) Nagarkot (hilltop) observed during 2013-2014, and (d) Pulchowk (urban) observed during November 2003-October 2004 in the Kathmandu Valley. Black dotted line represents WHO guideline (50 ppb) for daily maximum 8-hour average of O₃.

**Fig. 6.**

[Figure]

[Figure]

**Figure 7.** Diurnal pattern of hourly average O₃ mixing ratio for different seasons during January 2013-January 2014 at (a) Bode, (b) Paknajol, and (c) Nagarkot in the Kathmandu Valley. The four seasons (described in the text) are defined as: pre-monsoon (Mar-May), monsoon (Jun-Sep), post-monsoon (Oct-Nov), winter (Dec-Feb).

**Fig. 7.**

[Figure]

**Figure 8.** The estimated monthly average CO emission flux, which is based on the mean diurnal cycle of CO mixing ratios of each month for two conditions: (i) with data of all days (CO Flux) (blue dot) with lower and upper ends of the bar representing 25[th] and 75[th] percentile respectively, and (ii) with data of morning hours (CO Flux minimum (green dot) in which zero emission is assumed for the other hours of the day. The fluxes for July were not estimated as there were insufficient (less than 15 days) of concurrent CO and mixing layer height data. It is expected that the $F_{CO}$ and $F_{COmin.}$ for July should fall between values for June and August 2013.

**Fig. 8.**

**Supplement:**

**Response to reviewers' comments on Mahata et al. 2018**

We would like to thank the anonymous reviewers for their comments and suggestions which, we believe, have supported to improve the quality of the current manuscript. We have tried our best to incorporate both reviewers' comments in the manuscript. In the following responses, the reviewers' original comments are in black, authors' responses in blue and changes in the manuscript in red.

**Anonymous Referee #1**

**General remarks**

This paper reports air pollution (ozone and CO) in the Kathmandu area over longer time periods than hitherto available. Air pollution in this region is an important problem and reliable information covering all seasons is an important contribution to research on these issues. And I agree with the authors that the high ozone mixing ratios observed during the pre-monsoon period is of a high concern for human health and ecosystems, in the region. Here I would encourage the authors to go beyond what is presented in the paper and (briefly) discuss possible mitigation options (following the idea of "policy relevant, not policy prescriptive").

However, I have also some reservations about the interpretation of some aspects of the reported data and also some issues with the presentation. I suggest taking these points into account when revising the paper. If this is done in an appropriate way, I suggest that the Editor accepts the paper for publication in ACP.

We would like to thank you for considering that our study is of high importance for the region. We have tried our best to incorporate the suggestions in the revised manuscript.

**Comments in detail**

One aspect that is only discussed in passing in the paper is the role of stratospheric intrusions as a source of ozone in the upper troposphere in the region (e.g., Wang et al., 2012). Thus, ozone at higher altitudes in the troposphere could be enhanced independent of tropospheric pollution. I suggest that this aspect should be better discussed in the paper.

Thank you for noting the importance of stratospheric intrusions in the troposphere. We have included this fact in the manuscript in lines 524-538.

The diurnal profiles of $O_3$ mixing ratios (Figure 7) at three sites Bode and Pakanajol in the Valley and Nagarkot, a hilltop site normally above the Kathmandu Valley's boundary layer shows, notably in the morning hours, that the residual layer above the Kathmandu Valley's mixing layer contains a significant amount of ozone. Based on the surface ozone data collected at Paknajol during 2013-14, Putero e al. (2015) concluded that downward mixing of ozone from the residual layer contributes to surface ozone in the Kathmandu Valley in the afternoon hours (11:00-17:00 local time). It is likely that the same source has also contributed to higher ozone mixing ratios at Nagarkot. Such mixing has been observed at other sites as well. Wang et al. (2012) reported that the increase in downward mixing of $O_3$ from the stratosphere to the middle troposphere (56%) and the lower troposphere (13%) in spring and summer in Beijing. The downward flux was highest in the middle troposphere (75%) in winter. Similarly, Kumar et al. (2010) reported more than 10 ppb of stratospheric contribution to surface ozone at a high altitude site (in Nainital) during January to April. However, there were no significant stratospheric intrusions seen in spring and summer (seen only in winter) at Nepal Climate Observatry - Pyramid (NCO-P) located near the basecamp of Mt. Everest (Putero et al., 2016).

And discuss more about it in lines 547-557.

A study by Putero et al., (2015), based on $O_3$ mixing ratio measurements at Paknajol in the Kathmandu Valley, as a part of the SusKat-ABC campaign, has reported that the dynamics (both by horizontal and vertical winds) plays a key role in increased $O_3$ mixing ratios in the afternoon in the Kathmandu Valley. They estimated that the contribution of photochemistry varied as a function of the hour of the day, ranging from 6 to 34 %. Unfortunately, no viable NOx measurements were obtained at any site in the Kathmandu Valley and surrounding mountain ridges during the SusKat-ABC campaign. Speciated VOCs were measured at Bode only for about 2 months but NOx was not available for the same period. Therefore we were not able to discern quantitatively proportional contributions of NOx, VOCs and intrusion (chemistry vs. dynamics) from the free troposphere or lower stratosphere to observed $O_3$ concentrations at Nagarkot, Bode and other sites in the Valley.

Further, I suggest more comparison of the ozone pollution found at the Kathmandu valley with pollution levels elsewhere in the world (e.g. Huszar et al., 2016). Are the close to zero ozone values reported here (due to NO titration) also found in other regions of the world? These questions are important for mitigation strategies, because to achieve significant ozone reduction over cities in central Europe, the emission control strategies have to focus on the reduction of VOCs (Huszar et al., 2016).

Thanks for the suggestions. We have compared the level of $O_3$ observed in this study with the values reported in studies at other sites in different parts of the world. The new texts are in lines 498-503.

Similar patterns of ozone mixing ratios were observed at other sites in northern South Asia. For example, higher $O_3$ mixing ratios were observed in the afternoon (84 ppb) and lower during the night and early morning hours (10 ppb) at Kullu Valley, a semi-urban site located at 1154 m asl, in the North-western Himalaya in India (Sharma et al. 2012). A similar dip in $O_3$ value in the dark hours was observed at Ahmedabad, India by Lal et al. (2000).

And in lines 560-565.

However, air quality management plans need to consider carefully the reduction strategies of NMVOCs or NOx while aiming at mitigating the $O_3$ pollution in the Kathmandu Valley. If the correct strategy (NMVOCs vs. NOx) is not applied, then $O_3$ mixing ratios could increase, for example, as seen in Huszar et al. (2016) where they report that reducing NMVOCs in urban areas in central Europe leads to $O_3$ reduction whereas the focus on NOx reduction results in $O_3$ increase.

I repeat my comment on Fig. 1 from the initial/quick review here: I find the Google Earth figure not appropriate. The yellow pins are strange and the blue letters are difficult to read against the background. I suggest changing to a figure showing the locations of the sites in a map showing the orography clearly.

Thank you for the suggestion to improve the quality of Fig. 1. We have replaced the Google Earth figure by new one which is showing the orography clearly in map.

[Figure]

**Figure 1.** Observation sites in the SusKat-ABC international air pollution campaign during 2013-2014 in the Kathmandu Valley. A1 = Bode, A3 = Paknajol, and A4 = Naikhandi were selected within the valley floor and A2 = Bhimdhunga and A5 = Nagarkot on the mountain ridge. Naikhandi site is also near the Bagmati River outlet. Past study sites, Bouddha (X1) and Pulchowk (X2), which are referred in the manuscript, are also shown in the Figure. Source: Google map.

I also suggest to state the calendar months, not just the seasons. This is done in l. 271, but it should also be stated in the introduction and in the abstract.

It has been included in the introduction and abstract according to the reviewer's suggestion in lines 47-48 and lines 140, 166-167 and 171-172.

The value of for the CO flux at Bode is given to three significant numbers, is this really appropriate? Do you have an error estimate for this number? I think this value is an important result from this study so it deserves some attention.

We would like to thank you for pointing to error estimates in estimates of CO emission fluxes. Also, thank you for considering the CO flux estimate as an important result of our study.

- We have corrected the numbers to show only one significant digit in the abstract, chapter 3.5 and conclusions in the manuscript.
- Uncertainty in the CO flux estimate is introduced by the measurement uncertainty of the instruments, both for the CO mixing ratios and the mixing layer height (MLH). These contribute to the estimated CO emission fluxes varying over a wide range. In order to emphasize this wide variability in the estimated CO emission fluxes, we now show in Figure 8, the mean, interquartiles, and the minimum values of CO flux under our assumption.

Finally, I could very well imagine that the data presented in this paper are of interest to other researchers as well. Therefore I suggest to add a comment on data availability to the paper.

It is now required in the ACP to have a data availability section. The data availability section is included in lines 800-803 in the manuscript.

Data Availability: The observational data collected for this study will be made public through the SusKat website of IASS. They are also available upon direct request sent to maheswar.rupakheti@iass-potsdam.de and khadak.mahata@iass-potsdam.de.

**Minor issues**
• l. 31: drop 'on'
Incorporated.

• l. 32: 'pollutants'
Incorporated.

• l. 37: add altitude for Naikhandi

Incorporated.

• l. 42: State 'how long' extended

The campaign was extended until March 2014. This is included in the manuscript in line 43.

• l. 46: state the calendar months, not everybody is familiar with these seasons.

Calendar months are included in lines 47- 48.

• l. 46/47: 'due to the emissions from brick kiln industries' How do you know? How much of this is speculation/hypothesis how much is really shown in the paper?

Thank you for your question. Reviewer # 2 has also asked the same question. Our arguments are based on the previous studies. Previous studies carried out at the Bode site during the SusKat-ABC campaign have attributed over a dozen brick kilns located near Bode as strong sources of BC and EC (Kim et al., 2015; Mues et al., 2017), NMVOCs (Sarkar et al, 2016; Sarkar et al., 2017), SO2 (Kiros et al., 2016) and CO (Mahata et al., 2017), and the enhanced concentrations were observed during nighttime and mornings when winds blew from east and southeast bringing emissions from the location of the brick kilns to the observation site. Thus, we have rephrased the sentence (in lines 46-51 in abstract) as follows to better articulate it, and also explained in the main text with reference to other studies.

Seasonally, CO was higher during pre-monsoon season (March-May) and winter (December-February) season than during monsoon season (June-September) and post-monsoon (October-November) season. This is primarily due to the emissions from brick industries, which are only operational during this period (January-April), as well as increased domestic heating during winter, and regional forest fires and agro-residue burning during the pre-monsoon season.

The information is added in the texts in lines 415- 420 as follows:

Previous studies carried out at the Bode site during the SusKat-ABC campaign have attributed over a dozen brick kilns located near Bode as strong sources of BC and EC (Kim et al., 2015; Mues et al., 2017), NMVOCs (Sarkar et al, 2016; Sarkar et al., 2017), SO2 (Kiros et al., 2016) and CO (Mahata et al., 2017), and the enhanced concentrations were observed during nighttime and mornings when winds blew from east and southeast bringing emissions from the location of the brick kilns to the observation site.

• l.50: in which way did the meteorology play a role?

Thank you for the question. The role of meteorology has been explained in the abstract (lines 54-57) adding sentences as follows:

The wind is calm and easterly in the shallow mixing layer, with a mixing layer height (MLH) of about 250 m, during the night and early morning. The MLH slowly increases after the sunrise and decreases in the afternoon. As a result, the westerly wind becomes active and reduces the mixing ratio during the day time.

• l. 52: 'Some influence' is a bit vague, can you be more specific here?

Our study and a companion study by Bhardwaj et al. (2017) have identified the influence of emissions outside the Kathmandu Valley on the increase in ozone concentrations in the valley. The sentence has been rephrased in lines 57-60 as follows:

Furthermore, there was evidence of an increase in the $O_3$ mixing ratios in the Kathmandu Valley as a result of emissions in the Indo-Gangetic Plain (IGP) region, particularly emissions from biomass burning, including agro-residue burning.

• l. 54: The value of 4.92 is given to three significant numbers, is this really appropriate? Do you have an error estimate for this number?

Thank you for the suggestion. We agree. Thus, we kept it in round figure as discussed above in general comments. For example, 4.92 is rounded off to 4.9 in line 62.

• l. 63: 'as well as': which effect dominates?

Unfortunately, no viable NOx measurements were obtained at any site in the Kathmandu Valley and surrounding mountain ridge during the SusKat-ABC campaign. Speciated VOCs were measured at Bode only for about 2 months, but NOx data was not available for the same period. Therefore we were not able to discern proportional contributions of NOx, VOCs, and intrusion from free troposphere or lower stratosphere to observed $O_3$ concentrations at Bode and other sites in and around the Valley. A study by Putero et al., (2015), based on $O_3$ measurement at Paknajol in the Kathmandu Valley, as a part of the SusKat-ABC campaign, has reported that the photochemistry plays a key role ( larger role than the dynamics) in surface $O_3$ enhancement before noon and, together with the photochemistry, the boundary layer dynamics (both horizontal and vertical winds) also plays a role in increasing the $O_3$ mixing ratios in the afternoon (11:00-17:00 local time) in the Kathmandu Valley. They estimated that the contribution of photochemistry varied as a function of the hour of the day, ranging from 6 to 34 %. Due to unavailability of data on NOx and VOCs, we did not estimate which effect (chemistry vs. dynamics) is dominant in this case. Thus, we only slightly rephrased the sentence in lines 71-72 as follows:

…. air at the high-altitude site, as also indicated by Putero et al., (2015) for the Paknajol site in the Kathmandu Valley, as well as….

This is further explained in the lines 603- 605.

….as well as entrainment of ozone due to dynamics (both intrusion of ozone rich free tropospheric air into the boundary layer, and regional scale horizontal transport of ozone), as explained in case of Paknajol by Putero et al. (2015).

• l. 65: on the basis of which assessment can you say 'due to'?

This sentence would require more detailed explanation than is appropriate for the abstract, thus we have deleted this sentence from the abstract.

• l. 80: one further impact of local pollution could also be convective uplift to tropopause altitudes and transport into the extra-topical stratosphere in the monsoon season (e.g. Tissier and Legras, 2016, and references therein).

Thank you for this suggestion, which has been included in the manuscript. Revised sentences are as follows in lines 92-95.

Similarly, pollutants are also uplifted to the tropopause by convective air masses and transported to the extratropical stratosphere during the monsoon season (Tissier and Legras., 2016; Lawrence and Lelieveld, 2010; Fueglistaler et al., 2009; Highwood and Hoskins, 1998).

• l. 93: 2017 → 2018 • l. 97: also toxic outdoors?

Incorporated in lines 113 and 120.

• l. 115: measured → reported measurements

Incorporated in line 138.

• l. 133: for the Kathmandu . . .

Incorporated in line 157.

• l. 167: O3

Incorporated in line 192.

• l. 227: define 'AWS'

Automatic weather station (AWS) has been defined in line 241.

• l. 286: due to a problem

Incorporated in line 324.

• l. 320, 321: How do you know?

Past studies references have been included in lines 366-368 in the manuscript.

The morning peak at Bode was influenced by nighttime accumulation of CO along with other pollutants from nearby brick kilns (Sarkar et al., 2016; Mahata et al., 2017; Mues et al., 2017) and recirculation of air from above (Panday and Prinn, 2009).

• l. 339: CO mixing ratios

Incorporated in line 391.

• l. 453: stratospheric intrusions are mentioned here but only in passing.

Thank you for the suggestion. We have been inserted few sentences to elaborate it a bit in lines 547-557.

A study by Putero et al., (2015), based on $O_3$ mixing ratio measurements at Paknajol in the Kathmandu Valley, as a part of the SusKat-ABC campaign, has reported that the dynamics (both by horizontal and vertical winds) plays a key role in increased $O_3$ mixing ratios in the afternoon in the Kathmandu Valley. Unfortunately, no viable NOx measurements were obtained at any site in the Kathmandu Valley and surrounding mountain ridges during the SusKat-ABC campaign. Speciated VOCs were measured at Bode only for about 2 months but NOx was not available for the same period. Therefore we were not able to discern quantitatively proportional contributions of NOx, VOCs and intrusion (chemistry vs. dynamics) from the free troposphere or lower stratosphere to observed $O_3$ mixing ratios at Nagarkot, Bode and other sites in the Valley.

• l. 472: make → draw

Incorporated in line 573.

• l. 506: give altitude of Nagarkot here. Also the statement here is a bit vague, can you be more quantitative here (instead of 'but is also').

Thank you for the suggestion. We have rephrased the sentence in lines 608-611 as follows:

The ozone mixing ratios are relatively constant throughout the day at Nagarkot (~1901 m asl), which, being a hilltop site, is largely representative of the lower free tropospheric regional pollution values,  however, it is also affected by ozone production from precursors transported from the Kathmandu Valley due to westerly winds during the afternoon hours.

• Mues et al. (2017); citation is missing

Missing citation has been included in the reference.

• l. 546: be specific what is meant with 'this'

This represents mass per unit area. It has been changed in line 657.

• l. 611: change to 'an observation connected to'

Incorporated in line 749.

• l. 617: drop 'the' • l. 622: episode days → episodes
Incorporated in lines 755 and 760.

• l. 632: are these ozone values typical for down-mixing?

As explained earlier, a study by Putero et al., (2015), based on $O_3$ measurement at Paknajol in the Kathmandu Valley, as a part of the SusKat-ABC campaign, has reported that together with the photochemistry, the boundary layer dynamics (both horizontal and vertical winds) also plays a role in increasing the $O_3$ mixing ratios in the afternoon (11:00-17:00 local time) in the Kathmandu Valley. They estimated that the contribution of photochemistry varied as a function of the hour of the day, ranging from 6 to 34 %. A companion study by Bhardwaj et al. (2018) as also indicated role of dynamics in ozone levels in the Kathmandu Valley. The values we have observed are typical for down mixing. We have now cited these two previous studies in lines 769-773.

The diurnal cycle showed evidence of photochemical production, larger scale advection of polluted air masses as well as possible down-mixing of O3 during the daytime, as also observed by Putero et al., (2015) at Paknajol, with the hourly mixing 632 ratio at the polluted site increasing from typically 5-20 ppb in the morning to an early afternoon peak of 60-120 ppb (Putero et al., 2015; Bhardwaj et al., 2018).

• l. 646-650: perhaps two sentences here

Agree. The long sentence has been broken down to two as follows in lines 786-790.

This points out the need for the development of updated comprehensive emission inventory databases for this region. The improved emission inventory is necessary to provide more accurate input data to model simulations to assess air pollution processes and mitigation options for the Kathmandu Valley and the broader surrounding region.

• l. 711: This paper is now accepted

Incorporated in lines 855-858 in the reference section.

• Figs. 5 and 7: can you show error bars in these figures?

Thank you for the suggestions on adding error bars in Figs. 5 and 7. It will add value in the figures. The error bars has included in revised Figures 5 and 7 as suggested.

**Anonymous Referee #2**

The manuscript "Observation and analysis of spatio-temporal characteristics of surface ozone and carbon monoxide at multiple sites in the Kathmandu Valley, Nepal" by Mahata and co-authors provides an analysis of CO and O3 measurements carried out at 4 sites in the Kathmandu valley during the course of one year. Due to this good data coverage the analysis allows for a more thorough analysis than previously possible and also provides some valuable CO emission estimates. The paper is well written and organised and after minor corrections suitable for publication in ACP.

We would like to thank you for considering our study valuable, and for providing constructive comments to improve the quality of our analysis. We have tried our best to address your comments and suggestions in the revised manuscript.

**Minor comments**

L46 and elsewhere: Here a strong statement is made about the significant contribution of brick kilns to the observed CO concentrations. However, there is little actual proof of this shown in the manuscript. This could be improved by indicating the location of the kilns in relation to the measurement locations and a more thorough analysis/description of the nighttime wind pattern. Both of which would allow for a more creditable source attribution. Since there were also other atmospheric tracers measured at Bode, couldn't one of them (e.g. SO2 also be used to support the kiln contribution?

We would like to thank you for the suggestion. We have included the following evidence to more concretely attribute the influence of brick kiln on CO mixing ratios.

- As you suggested, we have revised the Figure 1 by marking the locations of brick kilns near the sampling site.
- Other studies conducted at the Bode site during the SusKat-ABC campaign attributed nearby brick kilns as strong sources of BC and EC (Kim et al., 2015; Mues et al., 2017), NMVOCs (Sarkar et al, 2016; Sarkar et al., 2017), SO2 (Kiros et al., 2016) and CO (Mahata et al., 2017), and the enhanced concentrations were during nighttime and mornings when winds blew from east and southeast bringing emissions from the location of the brick kilns to the observation site.

The revised text reads in lines 415-420 as follows:

Previous studies carried out at the Bode site during the SusKat-ABC campaign have attributed over a dozen brick kilns located near Bode as strong sources of BC and EC (Kim et al., 2015; Mues et al., 2017), NMVOCs (Sarkar et al, 2016; Sarkar et al., 2017), SO2 (Kiros et al., 2016) and CO (Mahata et al., 2017), and the enhanced concentrations were observed during nighttime and mornings when winds blew from east and southeast bringing emissions from the location of the brick kilns to the observation site.

L50, 51: Please mention in which way meteorology played a key role.

The role of meteorology has been included in lines 54-57(see also response to comment L50 by the reviewer 1). The new text reads as follows:

The wind is calm and easterly in the shallow mixing layer, with a mixing layer height (MLH) of about 250 m, during the night and early morning. The MLH slowly increases after the sunrise and decreases in the afternoon. As a result, the westerly wind becomes active and reduces the mixing ratio during the day time.

L72: Please split this number into casualties due to indoor and outdoor pollution. The first number seems to be the more important one in the light of your study.

Thank you for the suggestion. The impact on premature death due to outdoor and indoor air pollution has been included in lines 83-85.

The latest WHO report shows that the indoor and outdoor air pollution are each responsible for about 4 million premature deaths every year (http://www.who.int/airpollution/en/).

L83ff: Please also mention the special topographical and meteorological conditions (poor ventilation) that characterize the basin and further deteriorate air quality.

We have included the following line and also provided a reference on ventilation in lines 100-105.

In Kathmandu topography also plays a major role: the bowl-shaped Kathmandu Valley is surrounded by tall mountains and only a handful of passes. Topography is a key factor in governing local circulations, where low MLH  (typically in the range 250 m to 1,500 m)  and calm winds, have been observed particularly during nights and mornings. This in turn results in poor ventilation (Mues et al., 2017). Overall, this is conducive to trapping air pollutants and the deterioration of air quality in the valley.

L96: "CO is a useful tracer of urban air pollution". In the light of large contributions to CO from forest fires and agricultural waste burning (discussed later in the text), you should mention this important source as well.

Thank you for highlighting other important sources of CO other than urban sources. We have included forest and agro-residue waste burning sources of CO in lines 117-119.

Forest fires and agro-residue burning in the IGP and foothills of the Himalaya are other important contributors of CO in the region (Mahata et al., 2017; Bhardwaj et al., 2017).

L122: How does the CO emission estimate by Shrestha et al. (2013) compare with your emission estimate? Please add to the discussion in Section 3.5.

Shrestha et al., (2013) estimated the amount of CO emitted by a fraction of the vehicle fleet in the Kathmandu Valley. They neither estimated the total CO emission from all sources nor the CO fluxes in the Kathmandu Valley. Hence we cannot compare our estimate, which is from all sources, with theirs, which is from only a fraction of vehicle fleet.

L206 and for following sites: Where was the inlet mounted? What is the total height above ground of the inlet? Repeat from table 2.

The suggestion has been incorporated.  The new text reads as follows in

The inlets of the CO and $O_3$ analyzers were mounted on the roof top of the temporary lab, 20 m above the ground level.

…. building and its inlet was 2 m above ground. An automatic…..

…. The inlet of the $O_3$ analyzer was placed 25 m above the ground.

The instruments were kept in a one-story building of the school and its inlet was 5 m above the ground. The AWS….

…Nagarkot Health Post and their inlets were 5 m above the ground. The AWS…

L254: These IR CO analyzers usually show a strong drift with lab temperature. Did you assure that lab temperatures varied as little as possible (AC) or did you use some additional drift correction? Once daily zero checks would probably not be sufficient. Can you rule out that part of the observed diurnal cycle of CO is due to instrument errors?

We agree with your concern regarding a drift in IR based CO analyzers due to lab temperature. We didn't use an AC to maintain the temperature. We tried to keep the fluctuation in room temperature as small as possible using fans and windows. The IR-based CO monitor was run simultaneously with a co-located cavity ring down spectrometry based CO analyzer (Picarro CO analyzer) for ~ 3 months. The correlation coefficient and slope between their data are 0.99 and 0.96 respectively. This indicates there was very small drift in IR-based CO values (refer Mahata et al., 2017 for details). Therefore, we do not need to apply a correction to the IR-based CO data, and we can be confident that the observed diurnal cycle is not due to the instrument errors in IR-based CO measurements. We included a line as follows to reinforce this point in lines 283-289.

An IR-based Thermo CO monitor (model 48i-TLE) was run simultaneously with a co-located cavity ring down spectrometry based Picarro CO analyzer for nearly 3 months. The correlation coefficient and slope between the two measurements were found to be 0.99 and 0.96, respectively (Mahata et al., 2017). This indicates that there was very little drift in the IR-based CO values due to room temperature change, within acceptable range (i.e., within the measurement uncertainties of the instruments). Therefore, we did not any apply correction in the IR-based CO data.

L258: What was the result of the span check? Did the instrument drift since the last span check?

Thank you for your suggestion to make clear about the span check and drift of the instrument. We have included more information about it in line 289 as follows

The IR-based CO instruments' span drifts were within a 5 % range.

L283: What are the given uncertainties? Standard deviation of hourly observations? Uncertainty of the mean?

We would like to thank you for pointing out the confusion about the given uncertainties which are the standard deviation of the hourly averaged data, which is now clarified in lines 319-321 as follows:

The CO mixing ratios (measured in parts per billion by volume, hereafter the unit is denoted as ppb) of hourly averaged data over the total observation periods at four sites and their standard deviation were: Bode (569.9 ± 383.5) ppb…..

L302: Can you mention a bit more about what is known about the kind and timing of trash burning? Are these small scale fires (individual households) or larger scale (communities/neighbourhood)? Are there any regulations on this kind of waste treatment? It is mentioned elsewhere that this happens at night? Why? Seems to be a rather simple process to tackle to improve overall air quality.

Thanks for the suggestion. The information on waste type and timing has been added in lines 342-348.

Other studies conducted during the SusKat-ABC campaign have identified garbage (household waste and yard waste) burning as a key source of various air pollutants, such as OC and EC (Kim et al., 2015), PAHs (Chen et al., 2015), and NMVOCs (Sarkar et al., 2016; Sarkar et al., 2017). Garbage burning is often done in small fires and quite sporadic, normally taking place in the evenings and mornings (partly chosen to avoid attention from the responsible authorities). The rate of waste (and also biomass) burning in the morning is higher in winter due to the use of the fires for providing warmth on colder days.

L335: "support turbulent vertical diffusion". Although this statement is absolutely true, this is already reflected by the deeper mixing layer during daytime. I suggest to reformulate in such a way that the reasons for a deeper mixing layer are given in the first sentence (heating of surface by incoming solar radiation and (secondary) higher horizontal wind speeds and turbulence production). Then only mention the flushing effect of the increased wind speeds in the second sentence. In the end, the increased horizontal wind speeds are caused by the growing mixing layer height as well, so buoyancy production of turbulence is the real cause for the increased ventilation of the surface layer, but the above discussions seems to be sufficient.

The paragraph is rephrased according to the suggestion in lines 383-388.

The MLH starts increasing after radiative heating of the surface by incoming solar radiation. The heating of the ground causes thermals to rise from the surface layer resulting in the entrainment of cleaner air from above the boundary layer leading to the dissolution of nocturnal stable boundary layer. Increasing wind speeds (4-6 m s$^{-1}$) wind speeds (4-6 m s$^{-1}$) during daytime also support turbulent vertical diffusion, as well as flushing of the pollution …..

Section 3.2.1: You could also comment on the distinctly different shapes of the nighttime increase at Bode and Bhimdhunga. Bode shows an almost linear increase, which may indicate continued emissions into the local stable boundary layer, whereas Bhimdhunga shows a more isolated peak during the morning transition phase. So it would indicate that slope winds bring part of the polluted valley boundary layer up to the pass even in the early morning, which seems well possible considering the east facing slope above which the site is located. The same influence can be seen in $O_3$ at the site.

We would like to thank you for pointing out nighttime distinct shape of increased CO at Bode and Bhimdhunga. We are agree with your argument that the linear increase of CO at Bode is because of continuous addition of CO emitted from continuous sources nearby, i.e., brick kilns, in the shallow boundary layer. The isolated peak found at Bhimdhunga in morning at a mountain ridge could be due to elevated polluted layer brought up to the site by the up slope winds that start once the east-facing slope is heated by the morning sun. Thus, we have rephrased the paragraph in lines 375-380 as follows:

This is mainly associated with the persistent emissions such as those from brick kilns, which are in close proximity to the Bode measurement site under the stable boundary layer. The isolated peak during the morning transition phase at Bhimdhunga could be due to an elevated polluted layer because of the slope wind (Panday et al., 2009).

L351f: The argument about decreased forest fires and agricultural waste burning should be clarified a bit. Up to this point in the manuscript one had the impression that most of the CO at Bode was due to the brick kilns. But now the big difference between the seasons is explained through the absence of forest fires, etc and the brick kilns are only mentioned at the very end. When and why do they actually stop production? Due to the precipitation in the monsoon season?

The brick kilns are operated seasonally, from January to April every year. They are shut down the summer monsoon rainy period (June-September). We have rephrased the sentence and clarify about the brick kiln closure in monsoon period in lines 402-404 as follows:

Because of the rainfall, the brick production activities are stopped in the valley (usually they are operational from January-May every year). Further, the rainfall also….

L360: Why is this apparent? Even if the kilns operate at night you should show that there is a direct link to the site in terms of advection direction? Isn't residential heating the more likely candidate?

We have removed the confusing word "apparent". Brick kilns are operated even in the night time. Thus, the calm easterly wind brings pollutants from nearby brick kilns to the site (refer supplementary Figure S2 in Mahata et al., 2017; Mues et al., 2018). We have incorporated the suggestion in lines 412-414 as follows:

The nighttime accumulation of CO in Bode during pre-monsoon and winter is due to the influence of nearby brick kilns (Mahata et al., 2017) because of the calm easterly wind (refer supplementary Figure S2 in Mahata et al., 2017).

L368: There is also a distinct shift in the morning peak visible at all 3 sites for the different seasons. Can you please comment on this? Probably it is just due to an earlier onset of the morning transition in Mar-Apr, but maybe changes in local emissions may play a role as well.

Thank you for pointing out the clear shift in the morning peak at all sites in different seasons. We are in agreement with you regarding ca. one hour shift in morning peak from pre-monsoon to winter. This is due to earlier onset of the activities due to an earlier sunrise during the pre-monsoon than in winter. However, one hour shift in morning peak between Bode and Bhimdhunga/Naikhandi in pre-monsoon and winter is associated with commencement of early local emission under the shallow boundary layer at Bode. One hour lag in the morning peak at Bhimdhunga and Naikhandi may be due to uplifting of polluted layer and transport of city pollutants to the site, respectively, that starts only after the nearby slops are heated by solar radiation. One new paragraph is included in lines 429-435 as follows:

The distinct shift in the morning peak was seen at all 3 sites by season. The one hour shift in the morning peak from the pre-monsoon to winter is due to an earlier onset of the morning transition. However, the one hour difference in the morning peak between Bode (pre-monsoon at 8:00; winter at 9:00) and Bhimdhunga/Naikhandi (pre-monsoon at 9:00; winter at 10:00) in the pre-monsoon and winter is associated with commencement of early local emissions under the shallow boundary layer at Bode. The one hour lag in the morning peak at Bhimdhunga and Naikhandi may be due to transport of city pollutants to the site, respectively.

L393 "data means": Not clear which parameter is referred to here. Mean CO for the whole period or at a specific time of day?

Thank you for your suggestion to make it clear. We have tried to make it clear in lines 460-462.Yes, it is the mean CO of hourly data of the whole period. The rephrased sentences read as follows:

The t-test of the two hourly data means of CO in period I and period II at Bode, Bhimdhunga and Naikhandi (as in Figure 5) were performed at 95% confidence level and the differences were found to be statistically significant ($p < 0.5$).

L473: The comparability of the old time series with the recent may also be hampered by the difference in location and sampling height as well as a general difference in instrument calibration. These points should be mentioned as well.

Thank you for the suggestions. We have included them in the manuscript in lines 573-575.

…we cannot draw any conclusions about trends over the decade between the observations because of the difference in location and sampling height as well as a general difference in instrument calibration. However, a clear…

L499ff: Isn't the prolonged afternoon peak due to the same regional scale transport that was responsible for elevated CO? Free tropospheric contribution alone would not explain the difference between winter and pre-monsoon. Why not carry out the same kind of analysis as for CO in Figure 5.

We agree with you that the prolonged afternoon peak in the pre-monsoon could also be due to regional transport. We also carried out a similar analysis for $O_3$ (see figure below) as we did for CO in Figure 5, and also estimated the change in ozone mixing ratios (see table Table TS1) for the same two periods P1 and P2 (period 2 is being influenced by regional emissions). However, we don't see any prolonged afternoon peak on $O_3$ mixing ratios, at least not at Bode and Paknajol in the Valley, as we see in case of CO mixing ratios, likely because of the different reaction pathways of $O_3$ during night and day, and in different locations (valley floor vs. ridge)

Table. $O_3$ mixing ratios of period 1 (P1 = Mar 16-30) and period 2 (P2 = Apr 1-15) at Bode Paknajol and Nagarkot. The changes in $O_3$ mixing ratios and their percentage change at two periods were calculated during 24 hours and day time (8:00-18:00) hours.

| Site | P2-P1 24 hours | % change 24 hours | P2d-P1d 8:00-18:00 | % change in day hours 8:00-1800 |
|---|---|---|---|---|
| Bode | 13.4 | 33.8 | 13.7 | 26.2 |
| Paknajol | 12.7 | 31.6 | 19.1 | 35.0 |
| Nagarkot | 16.1 | 30.5 | 17.5 | 27.2 |

The % change in $O_3$ between two periods (P2-P1) is almost same at Bode and Nagarkot and highest at Paknajol in both 24 hours and day hours (8:00-18:00) calculations. It is likely that as explained earlier dynamics (horizontal transport, including regional transport and vertical down-mixing from the free troposphere) contributed to observed afternoon ozone mixing ratios, as pointed out by Putero et al. (2015). It is difficult to infer anything how much each process contributed to increase in $O_3$ values in period 2. In case of CO, it is clearly seen that the upwind sites (Bhimdhunga and Naikhandi), i.e, unwind to the Paknajol, Bode and Nagarkot during afternoon hours, had higher CO values than Bode in P2, indicating that there was a clear influence of regional transport of CO and hence led to prolonged afternoon peaks.

[Figure]

**Figure.** Comparison of hourly averaged $O_3$ mixing ratios during normal days (March 16-30) labelled as period I (dash line, faint color) and episode days (April 1-15) labelled as period II (line, dark color) in 2013 at Bode, Paknajol and Nagarkot.

Therfore, we have deleted the part that is poorly justified:
"Which Putero et al. (2015) suggested results in the broader afternoon peak of ozone during the pre-monsoon at Paknajol site, also observed at Bode site (and somewhat at Nagarkot)."
Therefore to reflect all the role of dynamics we have slightly modified the sentences in lines 601-605 as follows:

The typical $O_3$ maximum mixing ratio in the early afternoon at the urban and semi-urban sites is mainly due to daytime photochemical production as well as entrainment of ozone due to dynamics (both intrusion of ozone rich free tropospheric air into the boundary layer, and regional scale horizontal transport of ozone), as explained in case of Paknajol by Putero et al. (2015)

L505: The dip in O3 in the morning transition hours once more indicates the origin from the polluted stable boundary layer.

The suggestion has been included in the manuscript in lines 611-613

The dip in $O_3$ at Nagarkot (Figure 7) in the morning transition hours indicates the upward mixing of air from the polluted (and ozone-depleted) nocturnal boundary layer as it is breaking up.

L536 and equation 1: Why give t in hours? Why not just use seconds? Would save the conversion factor in the equation and is a better SI unit anyway!

This is because we used hourly averaged MLH and CO data. It is mentioned in the text (description of eq. 1).

542ff: One additional important limitation of the method is that of regional representativeness. As is said in the text, wind speeds are low so the observed CO increase at Bode may be rather localized and the emission estimate only valid for a small area and not for the whole city or valley. This is especially important when comparing the results with those from emission inventories that average over relatively large grid cells.

We agree with you that emissions are not uniform throughout the valley and thus our estimates may not be regionally representative. This can be checked only when we have a high resolution emission inventory, which (1 km x 1km spatial resolution) is being developed for the Kathmandu Valley and rest of Nepal (Sadavarte et. al., 2018). Therefore as per your suggestion, we have included a fifth point as one of our assumptions in lines 670-673.

v) CO emission is assumed to be uniform throughout the valley; this may not be correct, but cannot be verified until a high resolution emission inventory data is available, which is being developed for the Kathmandu Valley and rest of Nepal with a 1 km x 1km spatial resolution (Sadavarte et. al., 2018).

And we have revised assumption (iv) to make it clear in lines 660-663.

(iv) the vertical mixing of pollutants between the mixing layer and the free atmosphere is assumed to be negligible at night, thus strictly seen is the estimated CO flux calculated with eq. 1 only valid for the morning hours. When applied to the whole day the implicit assumption is that the emissions are similar during the rest of the 24 h period.

We have also reorganized 2$^{nd}$ paragraph of section 3.5 in lines 673-677.

During nighttime assumption (i) might not be entirely correct since the degree of mixing in the nocturnal stable layer and thus the vertically mixing is drastically reduced compared to daytime (and thus the term "mixing layer" is not entirely accurate, but we nevertheless apply it here due to its common use with ceilometer measurements). This adds a degree of uncertainty to the application of ceilometer observations to compute top-down emissions estimates, which will only be resolved once nocturnal vertical profile measurements are also available in order to characterize the nocturnal boundary layer characteristics and the degree to which the surface observations are representative of the mixing ratios throughout the vertical column of the nocturnal stable layer.

L555: Was the method actually applied to every night that had sufficient CO data? Or did you filter for low wind speed, constant MLH conditions? In which case it should be mentioned for how many nights per month the estimate was possible.

Yes, the method was applied when both CO and MLH hourly data were available during the night-morning hours. They were available for almost every day. Wind speed or any other filters were not applied during the flux calculation.

L574: Can you provide a realistic uncertainty for this estimate?

Thank you for the suggestion. Because the calculation of the emission flux is subject of several assumptions and there are no uncertainties given for the mixing layer height. It is not possible to calculate realistic uncertainties for the emission estimates. In order to still give an idea of the variability of the estimated flux numbers the 25$^{th}$ and 75$^{th}$ percentile is shown in Figure 8.

Instead of calculating realistic uncertainty, we have included few points (in lines 729-739) which have important role in flux estimates.

The emission estimates computed here are subject to several further uncertainties which are discussed in detail in Mues et al., (2017). In short, the uncertainties of CO flux estimates arise from (i) the assumptions that Bode site represents the whole atmospheric column and entire valley, which is not possible to verify without having many simultaneous monitoring stations in the valley (measurements at a few sites where CO was monitored for this study show some difference in CO mixing ratios), (ii) the higher variability (unclear minima and maxima during the morning and night hours) in the diurnal cycles of CO from June to October show a much higher variability than other months, that in turn makes it difficult to choose the exact hour of CO minimum and maximum needed for the flux estimation and (iii) the possible impact of wet deposition is not taken into account but would rather cause to generally underestimate the emission rate.

L587, the statement in brackets: Statement unclear? What do you mean by "averaged for the valley as a whole"? Did you apply the method also to other sites? Or just to Bode?

We have rephrased the sentence to avoid the confusion. We only apply this method at Bode. We do not have mixing layer height (MLH) measurements at other sites. The rephrased sentence reads in lines 713-716 as follows:

…..EDGAR HTAP V2.2 emission inventory database for 2010 [note that the CO emission values for the location at Bode and the whole valley (27.65-27.75°N, 85.25-85.40°E) were found to be the same up to two significant figures]….

L633: Again: mention the potential larger scale advection of polluted air masses (as for CO). See comment above.

The suggestion has been included in the conclusion section in lines 769-771. The rephrased sentence is as follows;

….The diurnal cycle showed evidence of photochemical production, larger scale advection of polluted air masses, as well as possible down-mixing of $O_3$ during the daytime, as also observed by Putero et al., (2015) at Paknajol, with the hourly….

Figure1: Instead of this 3D view, it would be more beneficial to have a plain 2D map with a scale indicator that would allow to identify the distances between sites. In addition, it would be a benefit to see the location of the large point sources (kilns) in such a map as well. Topography could still be included as isolines or shading. Main traffic routes would help as well.

Thank you for the suggestion. We have revised the figure as your suggestion. Please see reviewer 1's comments for the Figure 1above.

**References**

[revised manuscript text omitted]